# 63-year changes of annual streamflow volumes across Europe with a focus on the Mediterranean basin

Daniele Masseroni[1], Stefania Camici[2], Alessio Cislaghi[1,3], Giorgio Vacchiano[1,3], Christian Massari[2], Luca Brocca[2]

[1] Department of Agricultural and Environmental Sciences, University of Milan, Via Celoria 2, 20133 Milan, Italy
[2] Research Institute for Geo-Hydrological Protection, National Research Council, Via Madonna Alta 126, 06128 Perugia, Italy
[3] Centre of Applied Studies for the Sustainable Management and Protection of Mountain Areas (Ge.S.Di.Mont), University of Milan, Via Morino 8, Edolo, 25048 Brescia, Italy

*Correspondence to*: Daniele Masseroni (daniele.masseroni@unimi.it)

**Abstract.** Determining the spatio-temporal variability of annual streamflow volume plays a relevant role in hydrology for improving and implementing sustainable and resilient policies and practices of water resource management. This study investigates annual streamflow volume trends in a newly-assembled, consolidated and validated dataset of daily mean river flow records from more than 3,000 stations, which cover near-natural basins in more than 40 countries across Europe. Although the dataset contains streamflow time-series from 1900 to 2013 in some stations, the statistical analyses were carried out by including observations from 1950 to 2013 in order to have a consistent and reliable dataset over the continent. Trends were detected calculating the slope of Theil-Sen's line over the annual anomalies of streamflow volume.

The results show annual streamflow volume trends emerged at European scale, with a marked negative tendency in Mediterranean regions (about -1 $10^3$ m$^3$/(km$^2$ year)) and a generally positive trend in northern ones (about 0.5 $10^3$ m$^3$/(km$^2$ year)). The annual streamflow volume trend patterns appear in agreement with the continental-scale meteorological observations in response to climate change drivers. In the Mediterranean area, the declining of annual streamflow volumes started in 1965 and since early 80' volumes are consistently lower than the 1950-2013 average.

The spatio-temporal annual streamflow volume patterns observed in this work can help to contextualize short-term trends and regional studies already available in the scientific literature as well as to provide a valid benchmark for further accurate quantitative analysis on annual streamflow volumes.

## 1 Introduction

Elucidating continental patterns of annual streamflow volume changes in the Anthropocene epoch, to confirm unequivocally the effects of climate change and human impact on water resources, has become a challenge in contemporary hydrology (Bloschl et al. 2019). Although the hydrological scientific community undertook a great effort, few research robustly demonstrates an ubiquitous and uniform trend in European annual streamflow volumes (e.g., Mediero et al., 2015; Alfieri et al., 2015; Hodgkins et al. 2017; Blöschl et al. 2019). Most studies have identified two separate trends, both from recent

observations and using model projections sensitive to climate change: reduced flows in Southern and Eastern Europe (e.g., Stahl et al. 2010, Caloiero et al. 2018), and increased flows in Central and Northern Europe (up to minus or plus 45% after 1962 according to Teuling et al. 2019; -10-30% and +10-40% respectively by year 2050, under SRES A1B, according to Milly et al. 2005). Lehner et al. (2006)  indicated large critical regions in southern and southeastern Europe for which significant

changes in river flow drought are expected, and Feyen and Dankers (2009)  projected increases in streamflow drought severity and persistence in most parts of Europe.

Models have also highlighted a reduction through time of areas with increased runoff, and an expansion of those decreased runoff (e.g., a north-bound expansion of drying in the Mediterranean area) (Milly et al. 2005). Similar trends were also found when analyzing trends in zero flow days (Tremblay et al. 2020 ) and peak flows or flooding frequency, although with high

sensitivity to catchment size (Bertola et al. 2020 ). Most of such change has been attributed to changes in precipitation, with a less important role for land use and evapotranspiration change (Teuling et al. 2019).

Seasonal flows were also found to experience significant change (Bard et al. 2015, Bormann et al. 2017). Positive trends were found in the winter months in most catchments. A marked shift towards negative trends was observed in April, gradually spreading across Europe to reach a maximum extent in August. Low flows have decreased in most regions where the lowest

mean monthly flow occurs in summer, with some exceptions in catchments buffered by a large groundwater storage capacity (e.g. Fleig et al., 2010; Laizé et al., 2010). Bates et al. (2008) summarized European studies that have found generally similar but more spatially explicit patterns including, for example, decreasing future summer flow in Central and East Europe. Also, models sensitive to climate change project that the peak in discharge will occur approximately one month earlier due to increased temperatures and earlier snowmelt in the future, with changes that are much more pronounced and statistically

significant for all months under RCP8.5 compared to RCP4.5 (Lobanova et al. 2018).

Most studies, however, are based on observations limited to the second half of the 20th century (Piniewski et al. 2018, Renard et al. 2008, Birsan et al. 2005, KLIWA 2003, Schmocker-Fackel and Naef 2010, Demeterova and Skoda 2005, 2009, Fiala 2008, Fiala et al. 2010, Teuling et al. 2019).   In addition, several studies have highlighted the extreme sensitivity of river streamflow to data selection, method of trend detection, and time window for the analysis (Stahl et al. 2010). Kundzewicz et

al. (2005, 2017 ) advocated particular caution in interpreting streamflow trend signals resulting from a restricted number of stations with a small recording period, as even small gaps in the data time series or missing values could alter the significance of the statistical tests. Finally, even though trends highlighted by the literature are broadly consistent with spatial patterns of evapotranspiration and precipitation change, the effect of climate change on hydrology at the river basin scale is complex. Large-scale climate or hydrological models can reproduce broad patterns, but are still unable to capture all relevant spatially-

distributed characteristics of physical catchment structures and associated processes, particularly in regimes with storage and release of water across the seasons (Stahl et al. 2010). Also, extending the analysis to longer time series might reveal unexpected influences from long-term climate variability modes, such as the North Atlantic Oscillation (Hannaford et al. 2003 , Steirou et al. 2017 ) or expected changes in the Atlantic Meridional Overturning Circulation (Rousi et al. 2020 ), which might

introduce spurious trends in analyses focusing on shorter time spans. Finally, noise can be introduced by human modification

and appropriation of streamflow, which may also reverse forecasted changes in river flow (Forzieri et al. 2014).

On this background, it appears more useful soliciting large-scale studies in order to investigate predominant annual streamflow volume continental trends, which can provide basis for understanding processes in regional hydrology. In particular, a special effort should be carried out to focus the analysis of the river flow time series over the Mediterranean area, almost always overlooked in trend studies due to lack of river discharge data.

To achieve this aim, reliable networks of river streamflow measures in near-natural catchments are necessary. Several countries in the world have nowadays developed reference hydrometric networks composed by gauged stations with long and uninterrupted river flow records (Burn et al. 2012, Hannah et al. 2010). Such networks are generally managed and maintained by regional authorities or civil protection agencies, and are composed by gauging stations for measuring the river water level (stage) combined with updated stage-discharge relationships (Kundzewicz and Robson 2004). Datasets for large parts of

Europe are nowadays available (e.g. for Alpine, Mediterranean, Continental, Baltic and Nordic regions) with thousands of stations and records which starting from the nineteenth century. This amount of data covers a wide variety of catchments, from the small-size (few hundreds of hectares) to large-size (thousands of square kilometers) (Steiru et al. 2017, Mediero et al. 2014). Nevertheless, the development of the hydraulic infrastructures in Europe associated with the increase of population density and a lack of undisturbed natural environments makes measurement less representative of the natural flow conditions

(Bertola et al. 2019). Recording a high quality streamflow measurements unaffected by potential anthropogenic disturbances and suitable for large-scale trend analysis is a major challenge (Hisdal et al. 2001, 2007, Shorthouse and Arnell 1997).

The request of reference river flow dataset of near-pristine catchments has been largely recognized worldwide and has been supported by some international programs. The most famous is the FRIEND program, an initiative supported by the UNESCO International Hydrological Programme (IHO), the European Water Archive (EWA) and the European Environmental Agency

(EPA) that allows to share scientific information to improve methods applicable in water resources planning and management (Arnell 1997). However, updating streamflow measures and installing new flow meters is not straightforward in Europe. In particular, the organization has become complicated by regional and local jurisdictions, including political, administrative and technical constraints, as well as economical barriers (Viglione et al. 2010). In the absence of national or regional datasets, the Global Runoff Data Center (GRDC) can represent a valid global database of large continental river flow measures in Europe

(Haddeland et al. 2010, Stahl et al. 2010), despite most studies preferred to combine data with models predictions to fill gaps and reconstruct time series of comparable length (Dai et al. 2009). Hence, the challenge is combining the results of regional and national streamflow measures into a pan-European scale study of annual streamflow volume trends, which uses a consistent methodology on a consolidated and validated continental river flow dataset. In fact, detection and attribution of European trends in annual streamflow volumes can represent a strategic point in water management policies both in terms of flood

security, drought and desertification control (Ban et al. 2015). National and basin authorities could plan tailored irrigation methods in targeted areas as well as encourage the use of non-conventional water for irrigation or funding modernization of irrigation systems where streamflow negative trends occur (Rogger et al. 2017). On the contrary, authorities could promote

the use of natural water retention measures (http://nwrm.eu/) and best management practices (Urbonas and Stahre 1993) in territories affected by a positive trends (Brooks 2013).

The propose of the present study is, therefore, to provide an analysis of spatio-temporal variability of annual streamflow volumes in the European continent, starting from the analysis of consolidated observations over a long-time period with a particular emphasis on flow regimes relevant for water resource management especially in Mediterranean areas. Specifically, the added value of the present work is (i) to characterize annual streamflow volume trends over the entire European continent, using a long-time period of actual river flow observations, (ii) to deep the analysis on annual streamflow volume trends in the

Mediterranean area which is under increasing pressures of climate change effects, (iii) to determine whether evidences of a marked inversion point in the annual streamflow volume availability can be found directly in the observations, and (iv) to discuss the outcomes of the present study with previous investigations

## 2 Material and methods

### 2.1 River flow data selection and processes

A large dataset of daily river streamflow records measured by 3,913 gauged stations over the entire European continent was analysed for characterizing the continental patterns of the river flow regime over time. The original dataset, compiled by the authors, merges stations from 5 different databases, i.e., the Global Runoff Data Base (GRDC), the European Water Archive (EWA); the Italian ISPRA HIS national database (http://www.hiscentral.isprambiente.gov.it/hiscentral/default.aspx); the Portuguese national database (http://snirh.pt/) and the Spanish national database (http://ceh-

flumen64.cedex.es/anuarioaforos/default.asp), consisted of observed streamflow, recorded between 1900 and 2013. Unfortunately, not all the gauged stations have been worked since the same time (Fig. 1) and with a consistent and reliable dataset.

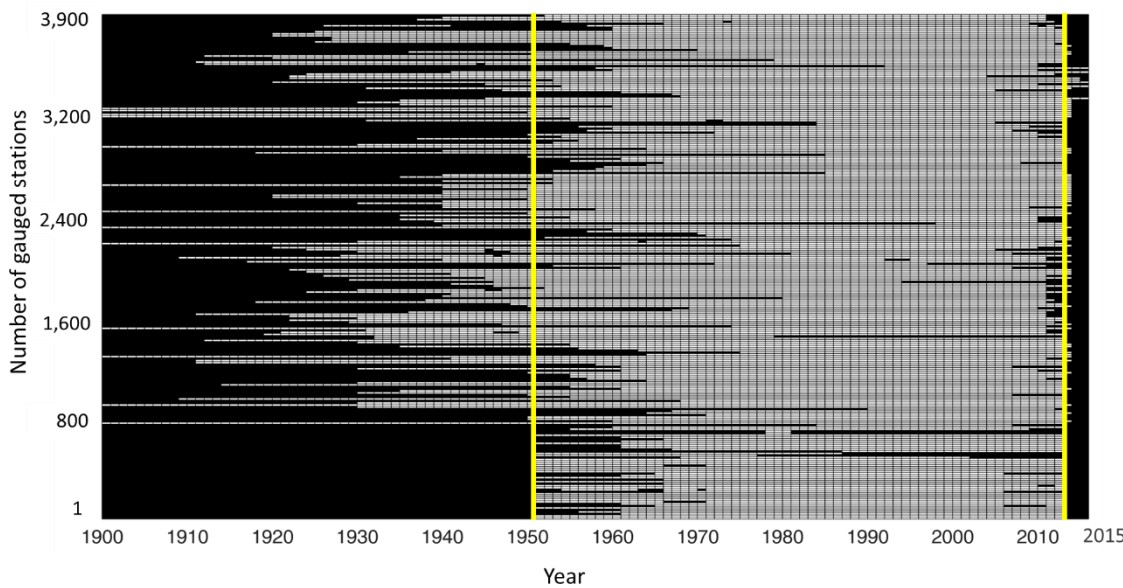

**Fig. 1. Available years for the 3,913 gauged stations. Vertical lines in yellow indicate the selected common study period 1950–2013.**

For assessing the reliability of streamflow daily values of each gauged station of the original dataset, a quality control and a homogeneity assessment were performed according the methodologies described in Buishand (1984), Chu et al. (2014), Ghiggi et al. (2019) and Kundzewicz (2015).

The quality control was conducted in succession on daily and aggregated time-series following the steps reported in Gudmundsson and Seneviratne (2016):

(i)  a visual hydrograph inspection to identify evident malfunction, consistent gaps and hydrograph disturbs such as presence of dams or reservoirs;

(ii)  excluding catchments with a drainage area larger than 100,000 km$^2$ to minimize the possibility that the human actives can significantly cause disturbances on the streamflow time-series (Piniewski et al. 2018);

(iii)  remove values with negative daily streamflow values;

(iv)  remove time-series with more than 2 years of missing data.

The homogeneity detection of data series was performed combining four different tests (Gudmundsson et al. 2018): (i) the standard normal homogeneity test of Alexandersson (1986); (ii) the Buishand range test (Buishand, 1982); (iii) the Pettitt test (Pettitt, 1979) and (iv) the Von Neumann ratio test (von Neumann, 1941). Homogeneity tests were carried out using the "iki.dataclim" statistical package for R (Orlowsky, 2014). The streamflow time series were considered as consistent when the null hypothesis at the 1% level was accepted at least in 3 of 4 tests (ECA&D) (Gudmundsson and Seneviratne, 2016; Merino et al., 2016).

Despite potential levels of human-induced alterations of river flow regime could be still present in time-series data after the application of the aforementioned controls, a certain degree of disturbance can be tolerated (Murphy et al. 2013). In order to further reduce the disturbance, high flow conditions were not investigated and we focused the analysis on annual streamflow volumes.

The application of quality control and homogeneity tests led to discard 428 series of data. Thus, 3,485 stations were selected to assembly a dataset that guarantees the best balance between the necessities to investigate a dataset as large as possible (which covers a large part of the continent and a nearly complete period of analysis), and to detect a historical variability. Location of the different gauged stations is reported in Fig. 2 on physical European map, whereas some statistics are reported in Tab.1. The selected gauged stations belong to more than 40 European countries especially over the Mediterranean basin. In fact, about one third are located in Spain, French and Italy. The dataset provided time-series data from 1950 to 2013 (i.e. 63-year study period has been considered as the maximum record length enable to guarantee a uniformity of series of data among the stations as reported in Fig. 1).

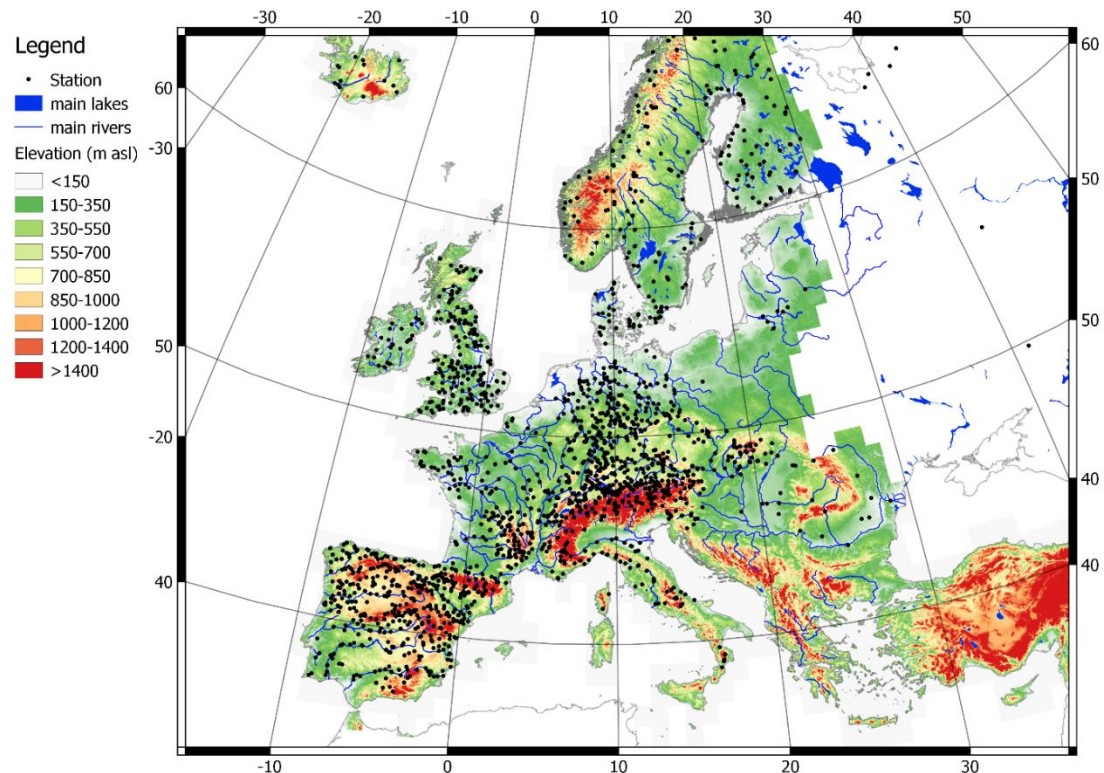

**Fig. 2. Map of European study area. Digital Elevation Model (DEM), main rivers and lakes as well as position of gauging stations are presented.**

**Tab. 1. Overall characteristics of examined basins.**

| Area range (km$^2$) | Percentage of basins (%) | Elevation of basin centroid (m asl) Maximum – Minimum (Mean) | Annual streamflow volume (Mln m$^3$) Maximum – Minimum (Mean) |
|---|---|---|---|
| 0-100 | 30 | 2900 - 2 (677) | 247.40 – 40.81 (112.78) |
| 100-200 | 21 | 2700 - 19 (510) | 241.85 – 44.15 (139.03) |
| 200-300 | 13 | 2170 – 30 (320) | 306.06 - 52.82 (154.01) |
| 300-400 | 10 | 2200 – 11 (621) | 338.43 – 68.38 (188.40) |
| 400-500 | 7 | 1980 – 10 (321) | 431.28 – 80.36 (246.83) |
| 500-600 | 6 | 1970 – 21 (452) | 526.43 – 106.32 (307.59) |
| 700-800 | 5 | 1856 – 31 (322) | 554.09 - 90.12 (312.32) |
| 800-900 | 3 | 1879 – 12 (398) | 671.32 - 98.89 (363.59) |
| 900-1,000 | 3 | 1900 – 10 (532) | 889.22 - 143.21 (488.03) |
| >1,000 | 2 | 1970 – 8 (601) | 931.21 - 150.01 ( 498.98) |

About 90% of stations belongs to catchments with size less than 1,000 km$^2$ of which more than 50% ranging from 1 to 200 km$^2$. Temporal autocorrelation level of the selected near-natural daily streamflow series was verified calculating lag-1 serial autocorrelation coefficient with a 95% of confidence bounds as suggested by Khaliq et al. (2009), Kulkarni and von Storch (1995) and von Storch (1995). All autocorrelation coefficients were found included in the confidence bounds, as shown in Fig. 3, and, therefore, they can be considered ready for the trend identification.

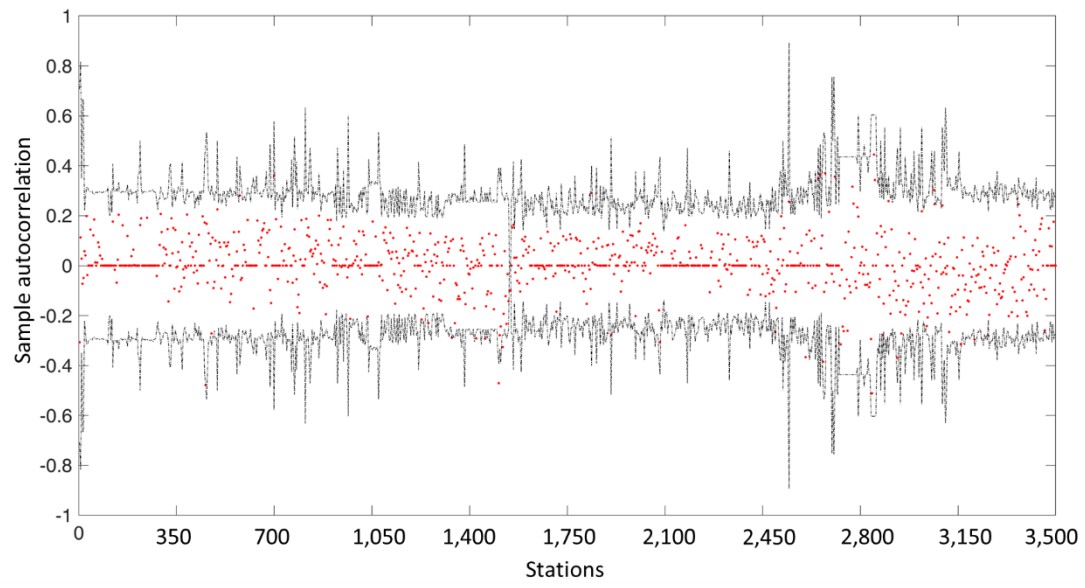

 **Fig.3. Samples autocorrelations. Red points are the value of lag-1 autocorrelation coefficient, whereas black dotted lines represent the 95% confidence bounds.**

## 1.2 Trend detection

Trend magnitude of a hydro-meteorological series of data is usually estimated using the Theil–Sen's estimator (Theil, 1950; Sen, 1968), a non-parametric test usually adopted for indicating monotonic trend and amplitude of change per unit time. It is a robust estimate of the magnitude of a trend in hydrological and climatic time-series as demonstrated in literature (e.g., Kundzewicz and Robson 2004, Stahl et al. 2010, Burn et al. 2012, Hammanfor et al. 2013). In the present study, the slope of Theil-Sen's line, known as Theil-Sen's slope or Sen's slope, was calculated on the annual anomalies in streamflow volumes, an innovative modality with respect to the application on direct streamflow data (Birsan et al. 2005). The annual anomalies in volumes were detected by comparing them with the baseline obtained by averaging annual streamflow volumes in the entire period of observation for each station. This strategy allows to emphasize trends, minimizing the random errors derived from uncorrected measures or unexpected signals, as already tested by Pandžić and Trninić (1992). A positive anomaly indicates that the observed annual streamflow volume is greater than the baseline, while a negative anomaly indicates the observed annual streamflow volume is lower than the baseline. The value of each anomaly was divided for the catchment area obtaining volume anomalies per unit of area. Moreover, significance of the annual streamflow volume trend was tested by adopting a non-parametric statistical approach based on Mann-Kendall (MK) (Mann, 1945; Kendall, 1975) test. Such test has already shown its robustness in trend detection, in particular in case of non-normally distributed data such as the meteorological and hydrological series (e.g., Yue and Wang 2002; Yue et al. 2003; Yue and Pilon 2004; Piniewski et al. 2018). In particular, if the result of the test is returned in H = 1, it indicates a rejection of the null hypothesis (i.e. presence of trend) at the alpha significance level (here assumed equal to 0.05). Conversely, if H = 0, it indicates a failure to reject the null hypothesis at the alpha significance level (i.e. no presence of trend). In the present study, we decided to maintain the integrity of the dataset focusing on the same time frame for all the study domain without splitting it in periods of different lengths. This procedure was already proposed in the study of Durocher et al. (2019) where preferred to discard all those time-series with missing data over a threshold rather than considered different time windows.

## 3 Results and discussion

### 3.1 Annual streamflow volume trends in Europe

Anomalies in annual streamflow volumes for each gauged station was calculated, and in Fig. 4a and b an example of positive and negative trend evaluated thought the slope of the Theil-Sen's line and confirmed by MK test for two stations located in central Europe, is reported.

a                                                                       b

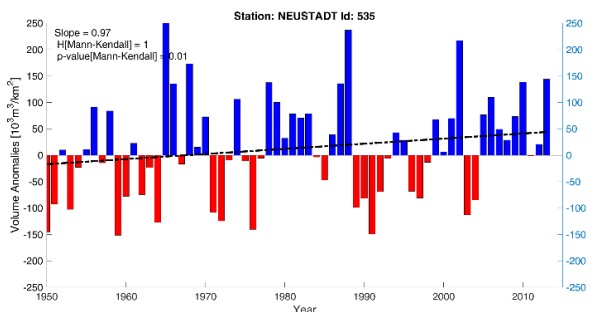 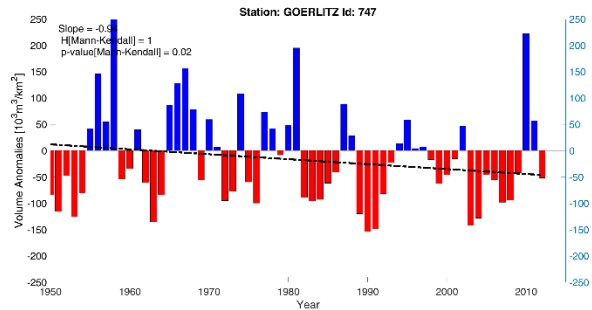

**Fig. 4. Anomalies in annual streamflow volumes for the station of Neustadt (a) and Goerlitz (b). In the graphs, the slope of the Theil-Sens's line, H and p-value of the Mann Kendall test are also reported.**

Results found that in 95% of the European gauged stations (i.e. 3,310 stations) the MK test confirmed the presence of a trend in annual streamflow volumes. In general, 70% of positive and 30% of negative trends in annual streamflow volume anomalies is recognized, with clear positive trend in northern regions and negative trend in southern ones, as shown in Fig. 5.

Adopting the subdivision of the European continent in the four macro-regions as provided by Gudmundsson et al. (2017) and Fernandez-Carrillo et al. (2019) i.e. Boreal, Continental, Atlantic and Mediterranean areas, the results show a marked negative trend in annual streamflow volumes especially in Mediterranean region with about 90% of stations with negative trend. The percentages of positive and negative trends for each macro-region are summarized in Tab. 2. The results reveal that, on average, a decrease in annual streamflow volume of about -1 $10^3$ m$^3$/(km$^2$ year) in Mediterranean areas and an increase of about 0.5 $10^3$ m$^3$/(km$^2$ year) in northern regions occur.

The spatial pattern of the annual streamflow volume trend reported in Fig. 5, appear broadly consistent with the findings obtained in previous sub-regional studies of Piniewski et al. (2018), Ilnicki et al. (2014), Bormann and Pinter (2017), Bard et al. (2015), Milly et al. (2005), Milliman et al. (2008), Manabe et al. (2004) and Dai et al. (2009). Although based on observed streamflow time-series with many differences (i.e., time interval, time length, methodology of measurement, etc.), sometimes affected by local river regulation or hydraulic infrastructure, and often completed with model-derived data, these studies predominantly found positive trends in regions close to the Atlantic Ocean and North Sea and negative trends in areas close to the Mediterranean Sea.

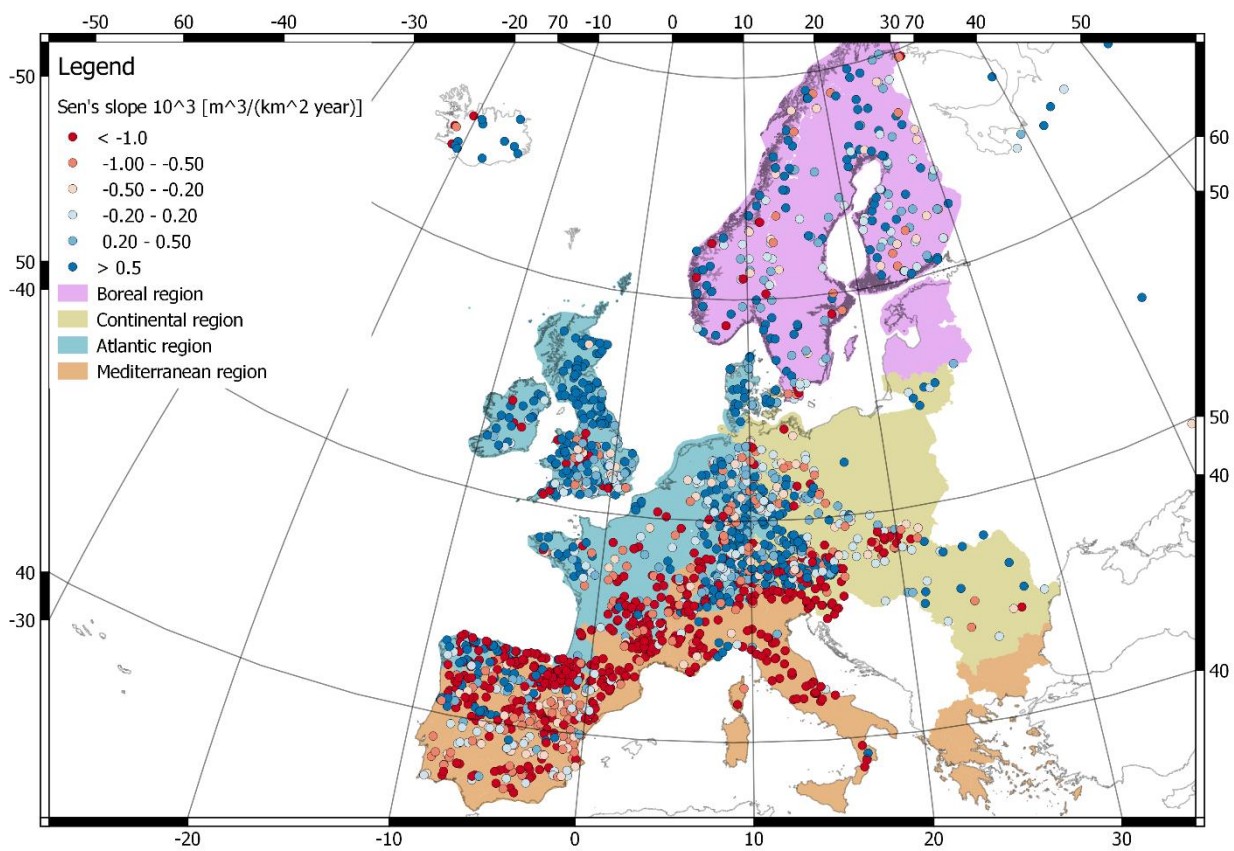

215

**Fig. 5. Annual trend of streamflow volume anomalies in European continent subdivided in Boreal, Continental, Atlantic and Mediterranean regions. Only significant trend are shown.**

**Tab. 2. Percentage of significant (i.e. 3,310 stations) positive and negative trends in annual streamflow volumes in the European**
220 **macro-regions.**

| Region | Number of stations | Positive trend | Negative trend |
|---|---|---|---|
| Boreal | 323 | 307 | 16 |
| Continental | 694 | 472 | 222 |
| Atlantic | 1191 | 846 | 345 |
| Mediterranean | 1102 | 88 | 1014 |
| Total | 3310 | 1713 | 1597 |

The European spatial pattern of the annual streamflow volume trend appears congruent also with the observed European rainfall long-period changes as shown in Fig. 6a, where the annual streamflow volume trends are overlapped to daily mean

rainfall trend maps obtained by E-OBS gridded dataset 20.0e (https://www.ecad.eu/ - Morice et al. 2012) for the same selected
225    period of daily streamflow series (i.e. 1950-2013).

A

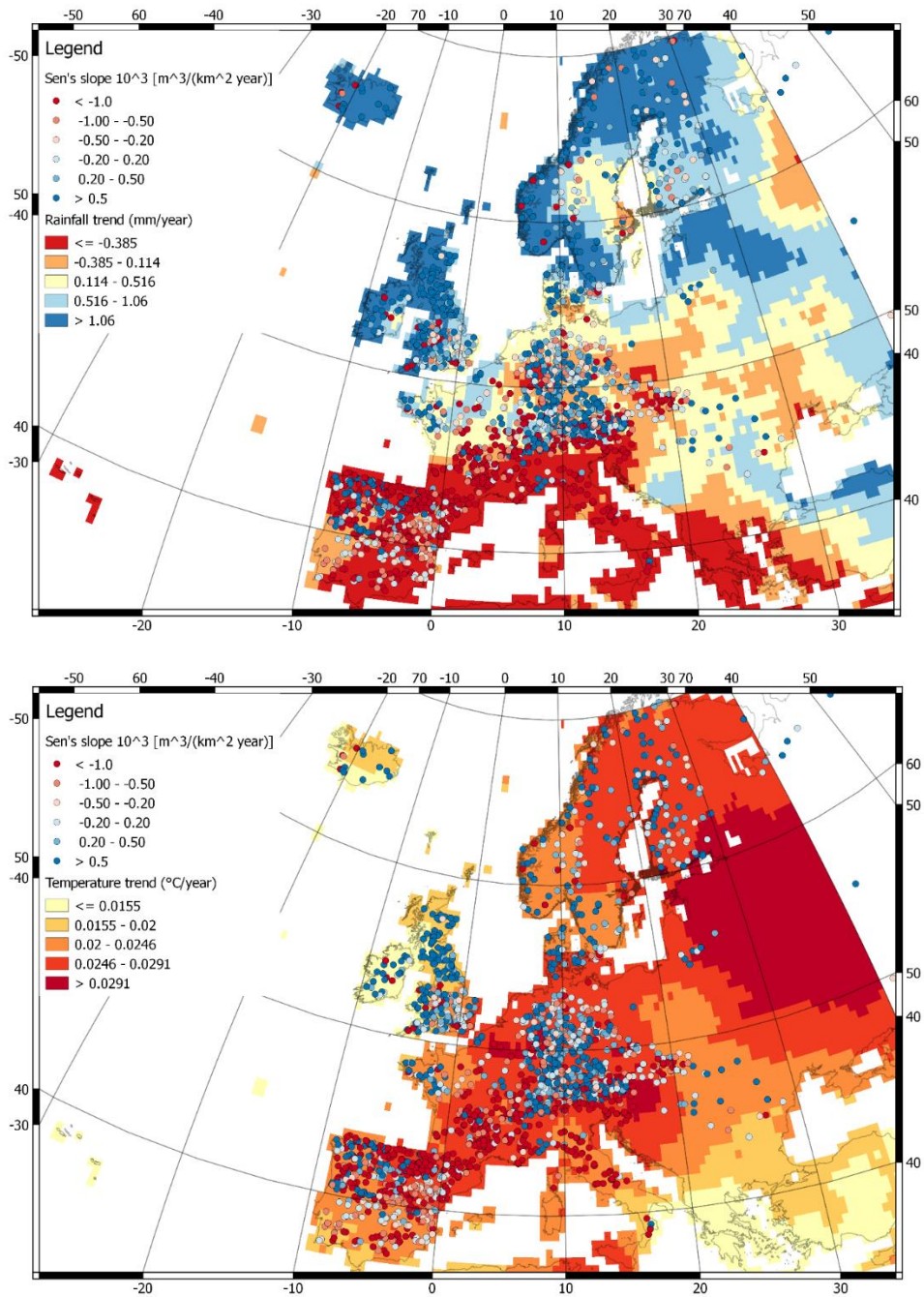

b

**Fig. 6. Comparison between annual streamflow volume trends and daily mean temperature (a) and rainfall (b) trends over the European continent. Only significant trend are shown.**

Concerning rainfall changes, the southern regions are affected by a marked negative trend (even below -3 mm/decade), while the northern regions are characterized by a positive trend which can overcome 10 mm/decade. The spatial distribution over the continent of both patterns appears perfectly congruent with the findings in annual streamflow volumes, as shown in Fig.6. Despite the spatial annual streamflow volume trend is very clear at a synoptic scale (i.e. increase of annual streamflow volumes in northern Europe and vice versa in southern Europe), in some local cases it can be opposite. In northern Germany, Scandinavian Peninsula and the east part of the Alps, positive and negative annual streamflow volume trends are mixed. This can be closely linked to complexity of snow-melt processes in glacier or mountain basins and the potential interactions between groundwater levels and river flows, as suggested by Renard et al. (2008), Birsan et al. (2005) and Pelliciotti et al. (2010). The authors found that in some regions such as southeast of England, northeast of France, as well as Danish the contribution of the aquifer to streamflow is high especially in summer periods. Various studies, moreover, have demonstrated that the mechanisms of interactions between groundwater and river flow contribute to moderate the influence of climate change drivers on streamflow, conversely, basins with less productive aquifers show a more direct response to climate drivers (Fleig et al. 2010, Laize et al. 2010).+

Concerning air temperature changes, the works of Staggle et al. (2017), Vicente-Serrano et al. (2014), Spinoni et al. (2015), Zeng et al. (2012), Willems (2013) and Madsen et al. (2014) confirm a global increase of mean temperatures with a marked trend in Mediterranean areas, where air temperature is expected to increase up to 0.3 °C/decade (as found in this study). The increase of air temperature directly impact glacierized and snow dominated basins where it can be responsible of the increase of runoff volume during the last sixty years due to the loss of ice masses (Sommer et al. 2020), however, depending on the basin elevation and trend in precipitation, some glaciers might have lost some sensitivity to an increased runoff production as a consequence of higher temperatures since there has not been more ice to melt and because, at high elevations, temperature might be not warm enough to counter balance the precipitation trend. In summary, for glacierized basins (or that use to be) there might be a causal effect of temperature on increased runoff volume (although this effect might have lost in time for some of them as explained below) while, for the others, precipitation seems again the main driver of runoff trend as it can be seen over the Alps by the contrasting trend found between the Italian side (negative) and continental side (positive) which reflects the trend in precipitation. On the other hand, temperature increase can impact negatively runoff over energy-limited environments by increasing evapotranspiration (Teuling et al. 2013, Avanzi et al. 2020) so some catchments might have experienced reduced runoff trend as a consequence of warming. This might explain the negative runoff trend found for some basins at high latitudes.

## 3.2 Annual streamflow volume trend in Mediterranean area

Focusing on the main Mediterranean river catchments (according with European Environmental Agency classification), the number of stations with positive and negative Theil-Sen's slope for each catchment was computed, and the results are reported in Fig. 7. In all main river basins in Spain, France and Italy prevails negative trends of annual streamflow volumes. The larger magnitude of negative annual streamflow volume trends is found in Garonne and Rhone river basins, respectively, of about -2.2 $10^3$ $m^3/(km^2$ year) and -3 $10^3$ $m^3/(km^2$ year). No basin with marked artefact trends is found as demonstrated by the very close distance of the 25th and 75th percentiles from the median slope value, confirming, thus, trend homogeneities inside each basin.

Negative trend over the entire Mediterranean basin is also confirmed by the analysis performed on the mean annual streamflow volume produced in this area. The annual streamflow volumes of each station were standardized by their mean value, and then the standardized annual streamflow volumes of all stations were averaged. The result is reported in Fig. 8, where the standardized annual streamflow volumes smoothed by a simple rolling average with a sliding window of 5-year length is shown along with the 25$^{th}$ and 75$^{th}$ percentile trends. When standardized annual streamflow volume is greater than 1 it means that the annual streamflow volume is greater than the average of annual streamflow volumes, vice versa if standardized annual streamflow volume is lower than 1. The former case can be considered as a positive signal of annual streamflow volume exceedance, whereas in the latter an annual streamflow volume deficit. Fig. 8 shows a change in the annual streamflow volume pattern between 1980 and 1985 moving from positive to negative availabilities with respect to the mean of annual streamflow volume observations. This finding is consistent with the results found by Hannaford et al. (2013) on the marked decreasing of low flow regimes in southern Europe in the last thirty years as well as with the conclusions of the International Panel of Climate Change (IPCC) work on climate change prospective (IPPC 2007) which highlighted how in the Northern Hemisphere climate change effects in reducing water resource availability have increased notably from the post- 1980 period.

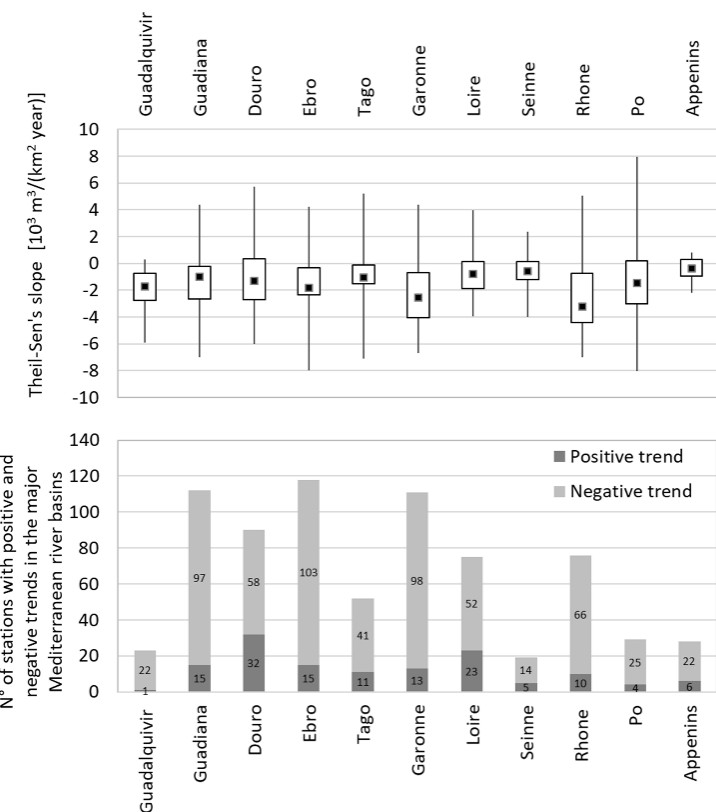

**Fig.7. Number of stations with positive and negative annual streamflow volume trends in the main Mediterranean river basins.**
**Box plot of the Theil-Sen's slope for each catchment is also reported.**

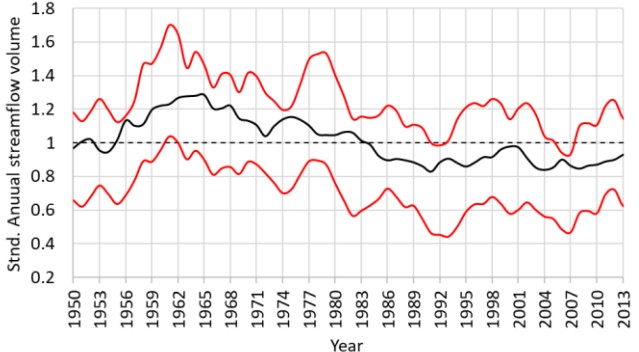

**Fig. 10. Standardized annual streamflow volume pattern from 1950 to 2013. Black line shows the simple rolling average of the standardized annual streamflow volume behavior over the entire Mediterranean area. Red lines show the 25th and 75th percentiles of the standardized annual streamflow volume series.**

## 4 Conclusions

This study closes the gap between regional researches on annual streamflow volume trend and a continental-scale pattern of its spatio-temporal variability. Starting from a dataset constituted by more than 3.000 gauge stations over more than 40 countries across Europe, anomalies in annual streamflow volume were computed and Theil-Sen's line slope was evaluated for each catchment over a recorded period from 1950 to 2013. A clear and undisputed trend pattern in annual streamflow volumes is recognized by the statistical analysis, showing marked negative trends in Mediterranean areas and positive trends in northern regions of Europe. All main Mediterranean river basins reveal negative trends in annual streamflow volume with an expected decreasing in annual streamflow volume of about $-1\ 10^3\ m^3/(km^2\ year)$. On the contrary, in northern regions of Europe, a positive increase of annual streamflow volume is expected to be on average about $0.5\ 10^3\ m^3/(km^2\ year)$. This trend patterns agree with the increase of temperatures and the decreasing in rainfall volumes detected by long-period observations on European continent. Indeed, these observations confirm an increase in drought situations in the southern regions of Europe, whereas revel an increase of rainfall volumes and runoff production in the northern European countries. In the Mediterranean area, the effect of climate change caused an inversion of the annual streamflow volume availability with respect to the mean of observations, i.e. from positive to negative values, starting from about 1985. In the recent 30-year period (1985-2013), the streamflow volumes are consistently lower than the average availability of the period 1950-2013.

The results of this study, therefore, can pave the way for more detailed quantitative analysis of annual streamflow volume variability (especially during different seasons) in order to meet the needs of managing water resources in agricultural, industrial and civil sectors.

## Acknowledgement

This study was developed in the context of IrriGate project "Toward a smart and flexible irrigation management in gravity-fed irrigation contexts' funded by Regione Lombardia PSR 1.2.01 year 2019.

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
