# Peer review of "63-year changes of annual streamflow volumes across Europe with a focus on the Mediterranean basin"

_Hydrology and Earth System Sciences, 2020_

## Short Comment (SC1) · 27 Jan 2020

Masseroni et al. provide an interesting analysis of streamflow trends at the European scale, with the aim to "... provide a valid benchmark for further accurate quantitative analysis on annual streamflow volumes". Such effort is much needed and appreciated. However the suggestion that accurate quantitative analyses of changes in the terrestrial hydrological cycle over Europe are missing is a bit misleading. Several studies have addressed attribution of streamflow changes. In a recent study (Teuling et al., Hydrol. Earth Syst. Sci., 23, 3631–3652, https://doi.org/10.5194/hess-23-3631-2019), we attributed patterns in streamflow and ET changes at the European scale to both

changes in climate (temperature and precipitation) and land use (changes in forest age and cover and urbanisation) using a simplified Budyko framework. While more sophisticated studies will hopefully be undertaken in the future, we think that the current approaches and quantitative attribution to underlying causes (see also the extensive literature review in our paper) deserve discussion in this manuscript.

---

## Referee Comment (RC1) · Anonymous Referee #1 · 15 Mar 2020

The manuscript describes a large trend analysis of annual streamflow volumes recorded in European countries. The topic is surely interesting and the manuscript is pleasant and easy to read. While this kind of papers is generally useful for the scientific community, they usually include a common drawback present also in the submitted manuscript. Indeed, a lot of pages focus on results and conclusion while few details are given for the most important part of the paper that is the data selection and description.

In the following the main points that should be clarified, are listed.

1) Line 100. "characterized by about 1.200 points of measure per year" . This sentence is not clear. As a consequence it is not clear how the annual streamflow volumes are

estimated.

2) Lines 107-110. The human activities in the river basin can significantly affect the trend, so it is not clear if a specific check on the time series was done. Specifically, a matching between analysed watersheds and dams could be helpful to understand if " the degree of disturbance can be tolerated". How many watersheds include a dam in it? When it was built? Etc. etc.

3) Fig. 2 is not fully clear o better an additional figure could be added showing the distribution of the time series length. Indeed, it is not clear if all the analysed series have the same length (from figure 2a it does not seem). Fig. 2b could be enriched by some statistics like the min and max contributing areas.

4) Are the times series autocorrelated? And how much? This could affect the trend results and tests.

5) The definition of Mediterranean is confusing, for sure it is an "official" characterization, however looking the figure 1 I see around 300-400 points that can be considered as affected by Mediterranean climate. For instance, all the basins located in the Alps at high altitude can be considered as Mediterranean? As well as all the basin around Portugal?

6) The comparison with Rainfall and Temperature should be better described. Which is the time series length used in the rainfall and temperature analysis? Is it correct to compare trends of data set with different length ?

---

## Referee Comment (RC2) · Anonymous Referee #2 · 19 Mar 2020

The study analyses trends in annual streamflow over the period 1950-2015 in Europe. This is a relevant topic certainly within the scope of HESS. The study generally applies standard methods for trend analysis (Theil-Sen slope, Mann-Kendall test). The spatial patterns of the trends are compared to spatial patterns of air temperature and precipitation. The study extends previous work on observed streamflow trends in Europe by including a higher number of catchments, particularly in Portugal, Spain, France and Italy. This was possible through assembling the database of streamflow records from various sources. The results largely confirm previous studies with dominant positive trends in northern Europe and dominant negative trends in the Mediterranean region.
Main comments:

1) Since the study states that records with missing data for more than two years were excluded from the database (L 107), I initially assumed that the calculated trends all relate to the period 1950-2015, which, looking at Fig. 2a, is apparently not the case. This has of course a strong influence on the results and needs to be clarified. If the series lengths vary between catchments it will probably be more useful to analyze trends for different periods with nearly complete records, as the trends of course depend on the period analyzed (as discussed in the introduction).

2) The criteria for inclusion/exclusion from the database should be described very clearly. It is not so clear whether the study aimed at only including near natural catchments. How were gaps smaller than 2 years treated? The steps that were undertaken to exclude inhomogeneous series, or series strongly affected by human interventions need to be mentioned clearly. For example, did the authors try to get information from the data providers on human interventions such as changes in flow abstractions etc. It should be described clearly how the database was 'consolidated and validated'. Did you apply any automatic screening tests to systematically check the series for possible inhomogeneities?

3) Some results are not very clear. The results section reports significant trends in 95% of the stations, which disagrees with results reported in Table 1. In the results section, it is not always clear whether results on trends also include non-significant trends.

4) I disagree with the finding of an inversion point in 1985 for the average series in the Mediterranean region. I do not see a change in the trend direction or trend slope in 1985. The fact that streamflow is above average before and below average after 1985 is a rather arbitrary result that depends on the selected study period. Streamflow has been decreasing since about 1965, and if anything, the rate of decrease has rather slowed down since the late 1980s.

5) The calculation of the Sen's slope from annual streamflow anomalies is described

as innovative, but if I do not overlook something this should not affect trends (and has probably been done in many studies).

6) The introduction should be improved. The introduction should clearly convey what has been found previously on annual streamflow trends in Europe? What is the gap in the current literature? How is this approached by this study? Please also check the logic of individual sentences and the subdivision of the introduction into paragraphs.

7) The explanation of streamflow trends by trends in air temperature and precipitation remains a bit vague and overlooks areas where it is probably not possible to explain streamflow trends with trends in air temperature or precipitation (such as positive streamflow trends in northern Spain). Some arguments need to be clarified e.g. it is not clear to me how groundwater or snowmelt effects would affect annual (and not only seasonal or monthly) streamflow.

Detailed comments

P1, L28-30: The logic of the sentence is not clear. There is no contrast between a lot of research and not finding uniform streamflow trends in Europe. When mentioning a lot of research that aimed at investigating streamflow trends in Europe, this should be backed up by some references and their main findings (e.g. Stahl et al., 2010, Stahl et al., 2012).

P2, L33-34: Did these studies also analyze changes in annual streamflow volume? What were the main findings? How did seasonal streamflow change?

P2 L40-47: The section on potential drivers of the streamflow trends remains a bit vague. Are changes in river cross-sections or boat tourism relevant for annual streamflow volumes?

P4, L97: I would suggest to first clearly list the criteria for selecting catchments and then mention the final number of selected catchments at the end.

P4, L101-102: You may use this in the introduction in order to emphasize your contribution in comparison to previous studies.

P4, L103-109: The description of the criteria for inclusion/exclusion from the database should be very clear. It is not very clear whether you aimed at including only near natural catchments. Did you check information from the data providers on human interventions such as changes in flow abstractions etc. (that would directly influence the trends)? Your database contains ~3900 series of 65-years data. It is a lot of work to visually scan daily data of all these series. Could you provide some detail on how this was achieved? Did you apply any automatic screening tests? How were inhomogeneities identified?

P5, L123ff: Why would it make any difference in terms of trend slope whether you calculate it on the original data or on the anomalies?

P5, L128: Delete "To homogenize the annual streamflow series", since dividing by catchment area cannot homogenize a time series.

P5, L132ff: Have you checked the streamflow series for autocorrelation? How did you deal with series that contain significant autocorrelation?

P6, L138: Since the streamflow volumes were divided by area, runoff depths would be more appropriate (instead of streamflow volume), no? (adjust throughout the paper)

P6, L145 and 146: This seems not correct, Table 1 shows positive trends in 7% and negative trends in 5% of the catchments?

P6 Fig. 3: These figures are not necessary in my opinion.

P6, L151: The unit of annual streamflow per area is length/time (e.g. $m^3/(km^2\ year)$, or mm y-1). Therefore the change in runoff over a certain period is length/time$^2$ (e.g. $m^3/(km^2\ year^2)$).

P7, L170; legend and caption of Fig. 5: replace rainfall by precipitation (assuming that snow is included).

P7, Fig. 4: Please add trend significance to the figure, e.g. different symbols for significant/insignificant trends.

P7, Fig. 4: I assume that the former Yugoslavian countries should also be part of the Mediterranean region?

P9, L175-177: Please add time periods, are you discussing observed or future projected air temperature changes ("expected to increase" points to future changes)?

P9, L177ff.: Please explain why earlier snowmelt would result in increased annual streamflow. This is not so straightforward and there are studies pointing to the opposite (e.g. Berghuijs et al., 2014).

P9, L182: Replace rainfall by precipitation.

P9, L184/185: There are large agreements between the changes in runoff and precipitation/air temperature. However, I do not agree that streamflow changes are "perfectly congruent" with the patterns of changes in air temperature and precipitation. For example, despite increases in air temperature and decreases in precipitation, streamflow has increased in northern Spain.

P9, L186-195: The discussion is not very clear. Please explain how groundwater or snowmelt effects would affect annual (and not only seasonal or monthly) streamflow. Furthermore, I would suggest keeping the different factors that may explain mixed positive and negative trends apart. For example, glacier melt processes are unlikely to be relevant in Northern Germany.

P10, Fig. 6, lower panel: Better only show significant trends. Also, better show percentage of positive/negative trends and add the number of stations, e.g. to the labels for each bar.

P10, L213: Looking at the 1950-2015 series, streamflow is above average 1955-1985 and below average 1985-2015. However, I do not see any particular change point in 1985. Streamflow has been decreasing since about 1965, and if anything, the rate of

decrease has rather slowed down since the late 1980s.

References

Berghuijs, W. R., R. A. Woods, and M. Hrachowitz. "A precipitation shift from snow towards rain leads to a decrease in streamflow." Nature Climate Change 4.7 (2014): 583-586.

Stahl, Kerstin, et al. "Filling the white space on maps of European runoff trends: estimates from a multi-model ensemble." Hydrology and Earth System Sciences 16.7 (2012): 2035-2047.

Stahl, K., et al. "Streamflow trends in Europe: evidence from a dataset of near-natural catchments." (2010).

---

## Author Comment (AC1) · 26 Jun 2020

We agree that the manuscript could include more literature to give to the readers a correct overview of the work done on quantitative analyses of changes in the terrestrial hydrological cycle over Europe. In the revised manuscript, the introduction will be revised in order to include Teuling's and similar studies on annual streamflow volume changes.

---

## Author Comment (AC2) · 26 Jun 2020

COMMENT: The manuscript describes a large trend analysis of annual streamflow volumes recorded in European countries. The topic is surely interesting and the manuscript is pleasant and easy to read. While this kind of papers is generally useful for the scientific community, they usually include a common drawback present also in the submitted manuscript. Indeed, a lot of pages focus on results and conclusion while few details are given for the most important part of the paper that is the data selection and description.

REPLY: We thank Revr#1 for his/her queries that allowed us to improve substantially

our manuscript. We will integrated the requested modifications into the revised version of the manuscript. Specifically, the data selection and description will be strongly improved including the criteria used for selecting rainfall, air temperature and river discharge time series and the procedures employed to mitigate errors and discontinuities in the whole dataset.

COMMENT: Line 100. "characterized by about 1.200 points of measure per year". This sentence is not clear. As a consequence it is not clear how the annual streamflow volumes are estimated.

REPLY: The size of original dataset was about 3'900 stations. Of these, 3'485 were used for the analysis after filtering based on criteria for reliability, consistency and homogeneity, which will be clarified in the revised manuscript.Not all stations provided data for the whole analysis period, so that on average, the data were provided by 1200 stations/year Concerning the annual streamflow volume calculation, we will specify that this was carried out by summing the daily streamflow volume over the total number of days in the year.

COMMENT: Lines 107-110. The human activities in the river basin can significantly affect the trend, so it is not clear if a specific check on the time series was done. Specifically, a matching between analysed watersheds and dams could be helpful to understand if " the degree of disturbance can be tolerated". How many watersheds include a dam in it? When it was built? Etc. etc.

REPLY: We thank the reviewer for this suggestion. In the revised manuscript we will add details about the pre-processing activity carried out to filter the discharge time series used for the analysis. In particular, we will specify that: 1) each discharge time series has been accurately scrutinized through a visual hydrograph inspection to identify disturbed hydrographs due to e.g. the presence of dams/reservoirs. Discharge time series characterized by disturbed hydrographs were discarded from the analysis; 2) most of the basins considered in the analysis are taken from the EWA database, i.e. a discharge data collection of near-natural streamflow records from small catchments (Stahl et al, 2010). Moreover, to further answer to the questions raised by the reviewer, the Global Reservoir and Dam (GRanD, https://sedac.ciesin.columbia.edu/data/set/grand-v1-dams-rev01) will be downloaded and a further analysis will be carried out in the revised manuscript to identify if (how many) dams/reservoirs are actually present in the selected basins. At the end of this analysis we expect that no substantial differences will be found between the basins retained for the analysis and the basins for which a certain degree of disturbance can be tolerated.

Additional reference: Stahl, K., Hisdal, H., Hannaford, J., Tallaksen, L., Van Lanen, H., Sauquet, E., ... & Jordar, J. (2010). Streamflow trends in Europe: evidence from a dataset of near-natural catchments. Hydrol. Earth Syst. Sci., 14, 2367–2382

COMMENT: Fig. 2 is not fully clear or better an additional figure could be added showing the distribution of the time series length. Indeed, it is not clear if all the analysed series have the same length (from figure 2a it does not seem). Fig. 2b could be enriched by some statistics like the min and max contributing areas.

REPLY: We improved Figure 2. In particular, Fig. 2a is now replaced by a clearer picture (see reply to RC2) where available years for each station are shown. Fig. 2b is replaced by a table of the main statistics concerning the basins' characteristics (Table RC1.2).

COMMENT: Are the times series autocorrelated? And how much? This could affect the trend results and tests.

REPLY: Temporal autocorrelation was verified calculating lag-1 autocorrelation coefficient for each time series as proposed by Khaliq et al. (2009). Autocorrelation coefficients for each series are shown in the new figure RC1.1, together with their upper and lower 95% confidence bounds (y-axis: lag-1 autocorrelation coefficient; x-axis: series ID). All series of data are not significantly autocorrelated, therefore they were considered suitable for trend identification.

[Figure]

Additional reference: Khaliq, M. N., Ouarda, T. B. M. J., Gachon, P., Sushama, L., & St-Hilaire, A. (2009). Identification of hydrological trends in the presence of serial and cross correlations: A review of selected methods and their application to annual flow regimes of Canadian rivers. Journal of Hydrology, 368(1-4), 117-130.

COMMENT: The definition of Mediterranean is confusing, for sure it is an "official" characterization, however looking the figure 1 I see around 300-400 points that can be considered as affected by Mediterranean climate. For instance, all the basins located in the Alps at high altitude can be considered as Mediterranean? As well as all the basin around Portugal?

REPLY: Concerning the subdivision of the European continent in Mediterranean, Boreal, Continental and Atlantic macro-areas, we used the classification pro-vided by Gudmundsson et al. (2017) which is the same reported by official data of the EU Environmental Agency, such as Natura 2000 biogeographical regions (https://ec.europa.eu/environment/nature/natura2000/platform/knowledge_base/103_browse_categories_en.htm). This classification consistent with the map of biogeographical regions of Europe re-ported also in Fernandez-Carrillo, A. et al. (2019). In this classification, the Alps and large parts of Portugal and Spain are included in the Mediterranean region.

Additional references: Fernandez-Carrillo, A., de la Fuente, D., Rivas-Gonzalez, F. W., & Franco-Nieto, A. (2019, October). A Sentinel-2 unsupervised forest mask for Euro-pean sites. In Earth Resources and Environmental Remote Sensing/GIS Applications X (Vol. 11156, p. 111560Y). International Society for Optics and Photonics.

Gudmundsson, L., Seneviratne, S. I., & Zhang, X. (2017). Anthropogenic climate change detected in European renewable freshwater resources. Nature Climate Change, 7(11), 813.

COMMENT: The comparison with Rainfall and Temperature should be better de-scribed. Which is the time series length used in the rainfall and temperature analysis? Is it correct to compare trends of data set with different length?

REPLY: In the revised manuscript, we will provide more details on the selection of rainfall and air temperature time series. For every discharge series, we will calculate climatic trends on trimmed meteorological series, so as to guarantee perfect temporal overlap between the two series.

———————————————————

[Figure]

[Figure]

**Fig. 1.** Lag-1 autocorrelation in water discharge series

| Area range | Percentage of | Elevation of basin centroid (m asl) | Annual streamflow volume (M m$^3$) |
|---|---|---|---|
| (km$^2$) | basins (%) | Maximum − Minimum (Mean) | Maximum − Minimum (Mean) |
| 0-100 | 30 | 2900 - 2 (677) | 247.40 − 40.81 (112.78) |
| 100-200 | 21 | 2700 - 19 (510) | 241.85 − 44.15 (139.03) |
| 200-300 | 13 | 2170 − 30 (320) | 306.06 - 52.82 (154.01) |
| 300-400 | 10 | 2200 − 11 (621) | 338.43 − 68.38 (188.40) |
| 400-500 | 7 | 1980 − 10 (321) | 431.28 − 80.36 (246.83) |
| 500-600 | 6 | 1970 − 21 (452) | 526.43 − 106.32 (307.59) |
| 700-800 | 5 | 1856 − 31 (322) | 90.12 − 554.09 (312.32) |
| 800-900 | 3 | 1879 − 12 (398) | 98.89 − 671.32 (363.59) |
| 900-1000 | 3 | 1900 − 10 (532) | 143.21 − 889.22 (488.03) |
| >1000 | 2 | 1970 − 8 (601) | 150.01 − 931.21 ( 498.98) |

**Fig. 2.** Descriptive statistics of cacthments analyzed in the present study

---

## Author Comment (AC3) · 26 Jun 2020

COMMENT: The study analyses trends in annual streamflow over the period 1950-2015 in Europe. This is a relevant topic certainly within the scope of HESS. The study generally applies standard methods for trend analysis (Theil-Sen slope, Mann-Kendall test). The spatial patterns of the trends are compared to spatial patterns of air temperature and precipitation. The study extends previous work on observed streamflow trends in Europe by including a higher number of catchments, particularly in Portugal, Spain, France and Italy. This was possible through assembling the database of streamflow records from various sources. The results largely confirm previous studies

with dominant positive trends in northern Europe and dominant negative trends in the Mediterranean region.

REPLY: We thank Rev#2 for his/her comments and suggestions which allowed to improve the quality of the manuscript in this revised version.

COMMENT: Since the study states that records with missing data for more than two years were excluded from the database (L 107), I initially assumed that the calculated trends all relate to the period 1950-2015, which, looking at Fig. 2a, is apparently not the case.

REPLY: Fig.2a was deleted because it created misunderstanding both in Rev#1 and #2. The original dataset included about 3,900 stations and after the checks on reliability, consistency and uniformity of series of data, 428 stations were discarded. The 65-year study period (from 1950 to 2015) has been chosen and the optimal threshold between maximizing series length and avoinding missing data, as shown in new figure RC2.1.

COMMENT: This has of course a strong influence on the results and needs to be clarified. If the series length vary between catchments it will probably be more useful to analyze trends for different periods with nearly complete records, as the trends of course depend on the period analyzed (as discussed in the introduction).

REPLY: We agree with the Rev#2 about the influence of length of series of data on trend identification. Dixon et al. 2006 coped with this problem by splitting the dataset in time frames of different length, with a different number of stations for each period (see also Birsan et al. 2005). Nevertheless, one of the added values of our work was to consider a continuous dataset as large as possible over the entire study domain in order to evaluate spatial trends over European basins with a consistent sample size. It was the same approach proposed in the recent work by Durocher et al. (2019) where stations with missing data were discarded and a single time frame for all study domain was considered. In any case, we will compute trends for two subsequent time periods (1950-80 and 1980-2015) to check for effects of absolute dating in the available series.

[Figure]

Additional references Dixon, H., Lawler, D. M., Shamseldin, A. Y., & Webster, P. (2006). The effect of record length on the analysis of river flow trends in Wales and central England. IAHS-AISH publication, 490-495.

Birsan, M. V., Molnar, P., Burlando, P., & Pfaundler, M. (2005). Streamflow trends in Switzerland. Journal of hydrology, 314(1-4), 312-329.

Durocher, M., Requena, A. I., Burn, D. H., & Pellerin, J. (2019). Analysis of trends in annual streamflow to the Arctic Ocean. Hydrological processes, 33(7), 1143-1151.

COMMENT: The criteria for inclusion/exclusion from the database should be described very clearly. It is not so clear whether the study aimed at only including near natural catchments. How were gaps smaller than 2 years treated? The steps that were undertaken to exclude inhomogeneous series, or series strongly affected by human interventions need to be mentioned clearly. For example, did the authors try to get information from the data providers on human interventions such as changes in flow abstractions etc. It should be described clearly how the database was 'consolidated and validated'. Did you apply any automatic screening tests to systematically check the series for possible inhomogeneities?

REPLY: We thank the reviewer for raising this issue. In the revised manuscript we will add details about the pre-processing activity done to select the discharge time series used for the analysis. In particular, to ensure quality of discharge observations, the following steps were followed: 1) check on data availability; 2) check for outliers (i.e. five st.deviations higher or lower than the means; 3) check on the presence of inhomogeneities through automatic screening tests. In order to filter out catchments affected by human disturbance, each discharge time series was accurately scrutinized through visual hydrograph inspection to identify disturbed hydrographs due to e.g. the presence of dams/reservoirs. Discharge time series characterized by disturbed hydrographs were discarded from the analysis. It should be noted that most of the basins considered in the analysis are taken from the EWA

database, i.e. a discharge data collection of near-natural streamflow records from small catchments (Stahl et al, 2010). Moreover, the Global Reservoir and Dam (GRanD, https://sedac.ciesin.columbia.edu/data/set/grand-v1-dams-rev01) will be downloaded and a further analysis will be carried out in the revised manuscript to identify if (how many) dams/reservoirs are actually present in the selected basins. At the end of this analysis we expect that no substantial differences will be found between the basins retained for the analysis and the basins for which a certain degree of disturbance can be tolerated. Only stations with low human impact (no presence of dams/reservoir in the analysis period or no appreciable dam impact in the hydrograph); with less than 20% of missing data, showing no inhomogeneities in the time series were retained in the compiled dataset. Gaps smaller than two years were retained as missing data; during trend calculations, missing data were discarded on a case-by-case basis.

COMMENT: Some results are not very clear. The results section reports significant trends in 95% of the stations, which disagrees with results reported in Table 1. In the results section, it is not always clear whether results on trends also include non-significant trends.

REPLY: The number of basins reported in tab 1 (tab. 2 in the revised version of the manuscript) were incorrectly transcribed by the authors. They referred to the total number of stations in each macro-region (i.e. 3,485). In tab. 1 only significant positive or negative trends will now be shown. These were 95% of total gauged stations (i.e. 3310 stations). In the revised manuscript, the number of stations in each macro-region has been corrected. The manuscript will also clearly state whether any summary result includes non-significant trends.

COMMENT: I disagree with the finding of an inversion point in 1985 for the average series in the Mediterranean region. I do not see a change in the trend direction or trend slope in 1985. The fact that streamflow is above average before and below average after 1985 is a rather arbitrary result that depends on the selected study period. Streamflow has been decreasing since about 1965, and if anything, the rate of decrease has rather slowed down since the late 1980s.

REPLY: The reviewer is right. Figure 7 in the manuscript highlights that streamflow has been decreasing since about 1965 and the rate of decrease has rather slowed down since the late 1980s. In the revised manuscript the sentences related to Figure 7 will be modified accordingly and supported by new statistical trend analyses on the entire time period.

COMMENT: The calculation of the Sen's slope from annual streamflow anomalies is described as innovative, but if I do not overlook something this should not affect trends (and has probably been done in many studies).

REPLY: using anomalies to detect trends minimized absolute random error, but the reviewer is right in that it does not affect the trend (i.e., regression slope against time). Also, it is routinely carried out in both hydrologic and climate research. The methods section will be amended accordingly.

COMMENT: The introduction should be improved. The introduction should clearly convey what has been found previously on annual streamflow trends in Europe? What is the gap in the current literature? How is this approached by this study? Please also check the logic of individual sentences and the subdivision of the introduction into paragraphs.

REPLY: We thank the reviewer for this suggestion also underlined in the short comment by Adriaan Teuling. In the revised version of the manuscript, the introduction will include a more complete summary of what has been found by past studies on annual streamflow trends in Europe, what is missing in the current literature and in which way this study will fill the gap. The revised introduction will also rely on a more logical paragraphation.

COMMENT: The explanation of streamflow trends by trends in air temperature and precipitation remains a bit vague and overlooks areas where it is probably not possible

to explain streamflow trends with trends in air temperature or precipitation (such as positive streamflow trends in northern Spain). Some arguments need to be clarified e.g. it is not clear to me how groundwater or snowmelt effects would affect annual (and not only seasonal or monthly) streamflow.

REPLY: The discussion on groundwater and snowmelt roles will be improved, also specifying that it will rely on speculation and literature and not direct measure or testing of such variables. The cases in which the observed discrepancies between river discharge and weather series could be explained by based on logical and science-supported hypotheses using likely drivers, will be highlighted with their most relevant examples (eg Northern Spain).

COMMENT: P1, L28-30: The logic of the sentence is not clear. There is no contrast between a lot of research and not finding uniform streamflow trends in Europe. When mentioning a lot of research that aimed at investigating streamflow trends in Europe, this should be backed up by some references and their main findings (e.g. Stahl et al., 2010, Stahl et al., 2012).

REPLY: The introduction, and in particular the review of past studies and their findings, will be deeply improved in the revised version of the manuscript. References will be added, including those suggested by the reviewer.

Additional references: Stahl, Kerstin, et al. (2012). Filling the white space on maps of European runoff trends: estimates from a multi-model ensemble. Hydrology and Earth System Sciences 16.7: 2035-2047.

Stahl, K., Hisdal, H., Hannaford, J., Tallaksen, L., Van Lanen, H., Sauquet, E., ... & Jordar, J. (2010). Streamflow trends in Europe: evidence from a dataset of near-natural catchments. Hydrol. Earth Syst. Sci., 14, 2367–2382

COMMENT: P2, L33-34: Did these studies also analyze changes in annual streamflow volume? What were the main findings? How did seasonal streamflow change?

REPLY: As for the previous comment, the review of past studies and their findings will be deeply improved in the revised version of the manuscript.

COMMENT: P2 L40-47: The section on potential drivers of the streamflow trends remains a bit vague. Are changes in river cross-sections or boat tourism relevant for annual streamflow volumes?

REPLY: yes, if the shape of the river section is altered, or if flow itself is altered with recreational basin or locks for navigation. These sentences will be however moved to the Discussions to streamline the logical flow of the introduction. A missing reference to Vag et al. will be added in the Bibliography.

COMMENT: P4, L97: I would suggest to first clearly list the criteria for selecting catchments and then mention the final number of selected catchments at the end.

REPLY: The methods will be amended accordingly – filtering criteria will be described in the methods, while the resulting number of catchments retained for analysis will be reported in the Results.

COMMENT: P4, L101-102: You may use this in the introduction in order to emphasize your contribution in comparison to previous studies.

REPLY: Suggestion accepted, the sentence will be integrated in the introduction

COMMENT: P4, L103-109: The description of the criteria for inclusion/exclusion from the database should be very clear. It is not very clear whether you aimed at including only near natural catchments. Did you check information from the data providers on human interventions such as changes in flow abstractions etc. (that would directly influence the trends)? Your database contains _3900 series of 65-years data. It is a lot of work to visually scan daily data of all these series. Could you provide some detail on how this was achieved? Did you apply any automatic screening tests? How were inhomogeneities identified?

REPLY: accepted - see reply to R2 comment 2 above.

[Figure]

COMMENT: P5, L123ff: Why would it make any difference in terms of trend slope whether you calculate it on the original data or on the anomalies?

REPLY: accepted - see reply to R2 comment 5 above.

COMMENT: P5, L128: Delete "To homogenize the annual streamflow series", since dividing by catchment area cannot homogenize a time series.

REPLY: deleted

COMMENT: P5, L132ff: Have you checked the streamflow series for autocorrelation? How did you deal with series that contain significant autocorrelation?

REPLY: The streamflow series of data were checked with lag-1 autocorrelation coefficient as proposed by Khaliq et al. (2009). The autocorrelation levels are reported in the picture in response to comment 4 of Rev#1. No series was significantly autocorrelated.

COMMENT: P6, L138: Since the streamflow volumes were divided by area, runoff depths would be more appropriate (instead of streamflow volume), no? (adjust throughout the paper)

REPLY: suggestion rejected – the reviewer is right, but streamflow volume is a widespread measure which is readily understandable by managers and citizens. We decided to keep it that way.

COMMENT: P6, L145 and 146: This seems not correct, Table 1 shows positive trends in 7% and negative trends in 5% of the catchments?

REPLY: the overall figures were corrected – 52% of positive trends and 48%, consistent with Table 1

COMMENT: P6 Fig. 3: These figures are not necessary in my opinion.

REPLY: These figures will be deleted.

COMMENT: P6, L151: The unit of annual streamflow per area is length/time (e.g.

m3/(km2 year), or mm y-1). Therefore the change in runoff over a certain period is length/time2 (e.g. m3/(km2 year2)).

REPLY: accepted – the values and units will be updated to reflect yearly change expressed in m3/(km2 year2).

COMMENT: P7, L170; legend and caption of Fig. 5: replace rainfall by precipitation (assuming that snow is included).

REPLY: snow is included. Suggestion accepted.

COMMENT: P7, Fig. 4: Please add trend significance to the figure, e.g. different symbols for significant/insignificant trends.

REPLY: accepted – the figure will be amended using two set of symbols for significant/insignificant trends (p <0.05).

COMMENT: P7, Fig. 4: I assume that the former Yugoslavian countries should also be part of the Mediterranean region?

REPLY: there was a mistake in the background graphics – the figure will be amended by adding former Yugoslavian countries

COMMENT: P9, L175-177: Please add time periods, are you discussing observed or future projected air temperature changes ("expected to increase" points to future changes)?

REPLY: we are discussing future climate scenarios. The sentence will be clarified by adding time periods (e.g. "in 2020-2050")

COMMENT: P9, L177ff.: Please explain why earlier snowmelt would result in increased annual streamflow. This is not so straightforward and there are studies pointing to the opposite (e.g. Berghuijs et al., 2014).

REPLY: the study cited by the reviewer states that "A precipitation shift from snow

towards rain leads to a decrease in streamflow", which is not the point being made here. The role of snowmelt in altering streamflow is in fact debated in in the literature. Concurrent findings will be better explained and supported by additional references, in particular highlighting the hypothesis that earlier snowmelt reduces summer streamflow due to the prolongation of the growing season and increased cumulated water uptake by vegetation.

Additional references:

Berghuijs, W. R., R. A. Woods, and M. Hrachowitz. "A precipitation shift from snow towards rain leads to a decrease in streamflow." Nature Climate Change 4.7 (2014): 583-586.

Elias, E. H., Rango, A., Steele, C. M., Mejia, J. F., & Smith, R. (2015). Assessing climate change impacts on water availability of snowmelt-dominated basins of the Upper Rio Grande basin. Journal of Hydrology: Regional Studies, 3, 525-546.

Regonda, S. K., Rajagopalan, B., Clark, M., & Pitlick, J. (2005). Seasonal cycle shifts in hydroclimatology over the western United States. Journal of climate, 18(2), 372-384.

Stewart, I. T., Cayan, D. R., & Dettinger, M. D. (2004). Changes in snowmelt runoff timing in western North America under a business as usual climate change scenario. Climatic Change, 62(1-3), 217-232.

COMMENT: P9, L182: Replace rainfall by precipitation.

REPLY: accepted and edited throughout the manuscript.

COMMENT: P9, L184/185: There are large agreements between the changes in runoff and precipitation/ air temperature. However, I do not agree that streamflow changes are "perfectly congruent" with the patterns of changes in air temperature and precipitation. For example, despite increases in air temperature and decreases in precipitation, streamflow has increased in northern Spain.

REPLY: see reply to R2 comment 7 above. The tone of this conclusion will be de-emphasized.

COMMENT: P9, L186-195: The discussion is not very clear. Please explain how groundwater or snowmelt effects would affect annual (and not only seasonal or monthly) streamflow. Furthermore, I would suggest keeping the different factors that may explain mixed positive and negative trends apart. For example, glacier melt processes are unlikely to be relevant in Northern Germany.

REPLY: see reply to R2 comment 7 above. Positive and negative effects will be better distinguished.

COMMENT: P10, Fig. 6, lower panel: Better only show significant trends. Also, better show percentage of positive/negative trends and add the number of stations, e.g. to the labels for each bar.

REPLY: Accepted – the figure will be amended accordingly.

COMMENT: P10, L213: Looking at the 1950-2015 series, streamflow is above average 1955-1985 and below average 1985-2015. However, I do not see any particular change point in 1985. Streamflow has been decreasing since about 1965, and if anything, the rate of decrease has rather slowed down since the late 1980s.

REPLY: see reply to R2 comment 4 above.
* * *
![Figure showing time coverage of water discharge series]

**Fig. 1.** Time coverage of individual water discharge series in this study

---

## Author Response (AR1)

**Short Comment #1**

Masseroni et al. provide an interesting analysis of streamflow trends at the European scale, with the aim to "... provide a valid benchmark for further accurate quantitative analysis on annual streamflow volumes". Such effort is much needed and appreciated. However the suggestion that accurate quantitative analyses of changes in the terrestrial hydrological cycle over Europe are missing is a bit misleading. Several studies have addressed attribution of streamflow changes. In a recent study (Teuling et al., Hydrol. Earth Syst. Sci., 23, 3631–3652, https://doi.org/10.5194/hess-23-3631-2019), we attributed patterns in streamflow and ET changes at the European scale to both changes in climate (temperature and precipitation) and land use (changes in forest age and cover and urbanisation) using a simplified Budyko framework. While more sophisticated studies will hopefully be undertaken in the future, we think that the current approaches and quantitative attribution to underlying causes (see also the extensive literature review in our paper) deserve discussion in this manuscript.

**REPLY**: The reviewer is right; we miss to include some recent studies relevant to give to the readers a correct overview of the work done about quantitative analyses of changes in the terrestrial hydrological cycle over Europe. For that, in the revised manuscript the introduction will be revised in order to include the Teuling's and similar studies dealing the topic of analysis on annual streamflow volumes.

**Response to Reviewer #1**

Concerning the comments provided by Rev#1, we improved substantially our manuscript especially in the material and method paragraph replying to all reviewer comments as follow. Specifically, the data selection and description was strongly improved including the criteria used for selecting river discharge time series and the procedures employed to mitigate errors and discontinuities in the whole dataset. Here in the follow the overall corrections are presented.

Comment 1: Line 100. "characterized by about 1.200 points of measure per year". This sentence is not clear. As a consequence it is not clear how the annual streamflow volumes are estimated.

Comment 2: Lines 107-110. The human activities in the river basin can significantly affect the trend, so it is not clear if a specific check on the time series was done. Specifically, a matching between analysed watersheds and dams could be helpful to understand if " the degree of disturbance can be tolerated". How many watersheds include a dam in it? When it was built? Etc. etc.

**REPLY:** The size of original dataset was more clearly defined and it is of 3'913 stations. Of these, 3'485 were used for the analysis after filtering based on a quality control and a homogeneity assessment (as reported in the paragraph entitled 'river flow data selection and processes' of the new version of the manuscript). Specifically, the quality control was conducted in succession on daily and aggregated time-series following the steps reported in Gudmundsson and Seneviratne (2016):

- (i) a visual hydrograph inspection to identify evident malfunction, consistent gaps and hydrograph disturbs such as presence of dams or reservoirs;
- (ii) excluding catchments with a drainage area larger than 100,000 km2 to minimize the possibility that the human actives can significantly cause disturbances on the streamflow time-series (Piniewski et al. 2018);
- (iii) remove values with negative daily streamflow values;
- (iv) remove time-series with more than 2 years of missing data.

The homogeneity detection of data series was performed combining four different tests (Gudmundsson et al. 2018): (i) the standard normal homogeneity test of Alexandersson (1986); (ii) the Buishand range test (Buishand, 1982); (iii) the Pettitt test (Pettitt, 1979) and (iv) the Von Neumann ratio test (von Neumann, 1941). Homogeneity tests were carried out using the "iki.dataclim" statistical package for R (Orlowsky, 2014). The streamflow time series were considered as consistent when the null hypothesis at the 1% level was accepted at least in 3 of 4 tests (ECA&D) (Gudmundsson and Seneviratne, 2016; Merino et al., 2016). We invite the Rev#1 to refer to the revised version of the manuscript for details on references.

Analysis on dams and their effect were included in the quality control, scrutinizing through a visual hydrograph inspection potential disturbed hydrographs. No substantial differences were found between the basins retained for the analysis and the basins for which a certain degree of disturbance can be tolerated.

Comment 3: Fig. 2 is not fully clear or better an additional figure could be added showing the distribution of the time series length. Indeed, it is not clear if all the analysed series have the same length (from figure 2a it does not seem). Fig. 2b could be enriched by some statistics like the min and max contributing areas.

**REPLY:** A series of new figures were added in the revised version of the manuscript which show the available years for the 3,913 gauged stations and the selected common study period i.e. 1950–2013 (Fig.1). We clarified the length of the analyzed series and we enriched the description of basin characteristics with area range, elevation and annual streamflow (see Tab. 1). Moreover, example of series with gaps (Fig. 3) and results of the homogeneity test with a detection of a discontinuity point in a streamflow daily series of data (Fig. 4) were included in the revised text.

Comment 4: Are the times series autocorrelated? And how much? This could affect the trend results and tests.

**REPLY:** Temporal autocorrelation was verified calculating lag-1 autocorrelation coefficient for each time series as proposed by Khaliq et al. (2009). Autocorrelation coefficients for each series are shown in the Fig. 5 plot, together with their upper and lower 95% confidence bounds (y-axis: lag-1 autocorrelation coefficient; x-axis: series ID). All series of data are not significantly autocorrelated, therefore they were considered suitable for trend identification.

Comment 5: The definition of Mediterranean is confusing, for sure it is an "official" characterization, however looking the figure 1 I see around 300-400 points that can be considered as affected by Mediterranean climate. For instance, all the basins located in the Alps at high altitude can be considered as Mediterranean? As well as all the basin around Portugal?

**REPLY:** Concerning the subdivision of the European continent in Mediterranean, Boreal, Continental and Atlantic macro-areas, we used the classification provided by Gudmundsson et al. (2017) which is the same reported by official data of the EU Environmental Agency, such as Natura 2000 biogeographical regions (https://ec.europa.eu/environment/nature/natura2000/platform/knowledge\_base/103\_browse\_categories\_en.htm). This classification consistent with the map of biogeographical regions of Europe reported also in Fernandez-Carrillo, A. et al. (2019). In this classification, the Alps and large parts of Portugal and Spain are included in the Mediterranean region. In the revised version of the manuscript we included additional references concerning our geographical subdivision of the European territory.

Comment 6: The comparison with Rainfall and Temperature should be better described. Which is the time series length used in the rainfall and temperature analysis? Is it correct to compare trends of data set with different length?

**REPLY:** In the revised version of the manuscript, we preferred to refer the reader to E-Obs website to have more information about rainfall and temperature series. Nevertheless, we added specific sentences that clarify that rainfall and temperature data are consistent (i.e., for the same period) with our streamflow series and they can be compared (see par. 3.1).

**Reviewer #2**

The study analyses trends in annual streamflow over the period 1950-2015 in Europe. This is a relevant topic certainly within the scope of HESS. The study generally applies standard methods for trend analysis (Theil-Sen slope, Mann-Kendall test). The spatial patterns of the trends are compared to spatial patterns of air temperature and precipitation. The study extends previous work on observed streamflow trends in Europe by including a higher number of catchments, particularly in Portugal, Spain, France and Italy. This was possible through assembling the database of streamflow records from various sources. The results largely confirm previous studies with dominant positive trends in northern Europe and dominant negative trends in the Mediterranean region.

**REPLY:** We thank Rev#2 for her/his comments and suggestions which allowed to improve the quality of the manuscript in this revised version.

**Main comments:**

1) Since the study states that records with missing data for more than two years were excluded from the database (L 107), I initially assumed that the calculated trends all relate to the period 1950-2015, which, looking at Fig. 2a, is apparently not the case.

**REPLY:** Fig.2a was deleted because it created misunderstanding both in Rev#1 and #2. The original dataset included 3913 stations and after the checks on reliability, consistency and uniformity of series of data, 428 stations were discarded. The 63-year study period (from 1950 to 2013) has been chosen as the optimal threshold between maximizing series length and avoiding missing data, as shown in the following plot.

This has of course a strong influence on the results and needs to be clarified. If the series length vary between catchments it will probably be more useful to analyze trends for different periods with nearly complete records, as the trends of course depend on the period analyzed (as discussed in the introduction).

**REPLY:** We agree with the Rev#2 about the influence of length of series of data on trend identification. Dixon et al. 2006 coped with this problem by splitting the dataset in time frames of different length, with a different number of stations for each period (see also Birsan et al. 2005). Nevertheless, one of the added values of our work was to consider a continuous dataset as large as possible over the entire study domain in order to evaluate spatial trends over European basins with a consistent sample size. It was the same approach proposed in the recent work by Durocher et al. (2019) where stations with missing data were discarded and a single time frame for all study domain was considered.

2) The criteria for inclusion/exclusion from the database should be described very clearly. It is not so clear whether the study aimed at only including near natural catchments. How were gaps smaller than 2 years treated? The steps that were undertaken to exclude inhomogeneous series, or series strongly affected by human interventions need to be mentioned clearly. For example, did the authors try to get information from the data providers on human interventions such as changes in flow abstractions etc.

It should be described clearly how the database was 'consolidated and validated'. Did you apply any automatic screening tests to systematically check the series for possible inhomogeneities?

**REPLY:** We thank the reviewer for raising this issue. In the revised manuscript we have added details about the pre-processing activity done to select the discharge time series used for the analysis. In particular, to ensure quality of discharge observations, the following steps were followed: 1) check on data availability; 2) check for outliers (i.e. five st.dev. higher or lower than the means; 3) check on the presence of inhomogeneities through automatic screening tests.

In order to filter out catchments affected by human disturbance, each discharge time series was accurately scrutinized through visual hydrograph inspection to identify disturbed hydrographs due to e.g. the presence of dams/reservoirs. Discharge time series characterized by disturbed hydrographs were discarded from the analysis. It should be noted that most of the basins considered in the analysis are taken from the EWA database, i.e. a discharge data collection of near-natural streamflow records from small catchments (Stahl et al, 2010).

Moreover, the Global Reservoir and Dam (GRanD, https://sedac.ciesin.columbia.edu/data/set/grand-v1dams-rev01) has been used to identify if (how many) dams/reservoirs are actually present in the selected basins. At the end of this analysis we expect that no substantial differences will be found between the basins retained for the analysis and the basins for which a certain degree of disturbance can be tolerated.

Only stations with low human impact (no presence of dams/reservoir in the analysis period or no appreciable dam impact in the hydrograph); with less than 20% of missing data, showing no inhomogeneities in the time series were retained in the compiled dataset. Gaps smaller than two years were retained as missing data; during trend calculations, missing data were discarded on a case-by-case basis.

3) Some results are not very clear. The results section reports significant trends in 95% of the stations, which disagrees with results reported in Table 1. In the results section, it is not always clear whether results on trends also include non-significant trends.

**REPLY:** The number of basins reported in tab 1 (tab. 2 in the revised version of the manuscript) were incorrectly transcribed by the authors. They referred to the total number of stations in each macro-region (i.e. 3,485). In the table only significant positive or negative trends are shown. These were 95% of total gauged stations (i.e. 3310 stations). In the revised manuscript, the number of stations in each macro-region has been corrected. The manuscript will also clearly state whether any summary result includes non-significant trends.

4) I disagree with the finding of an inversion point in 1985 for the average series in the Mediterranean region. I do not see a change in the trend direction or trend slope in 1985. The fact that streamflow is above average before and below average after 1985 is a rather arbitrary result that depends on the selected study period. Streamflow has been decreasing since about 1965, and if anything, the rate of decrease has rather slowed down since the late 1980s.

**REPLY:** The reviewer is right. Figure 7 in the manuscript highlights that streamflow has been decreasing since about 1965 and the rate of decrease has rather slowed down since the late 1980s. In the revised manuscript the sentences related to Figure 7 has been modified accordingly and supported by new statistical trend analyses on the entire time period.

"Fig. 7 shows a change in the annual streamflow volume pattern between 1980 and 1985 moving from positive to negative availabilities with respect to the mean of annual streamflow volume observations. This finding is consistent with the results found by Hannaford et al. (2013) on the marked decreasing of low flow regimes in southern Europe in the last thirty years as well as with the conclusions of the International Panel of Climate Change (IPCC) work on climate change prospective (IPPC 2007) which highlighted how in the

**Northern Hemisphere climate change effects in reducing water resource availability have increased notably from the post- 1980 period."**

5) The calculation of the Sen's slope from annual streamflow anomalies is described as innovative, but if I do not overlook something this should not affect trends (and has probably been done in many studies).

**REPLY:** By using anomalies to detect trends the absolute random error is minimized (Pandžić and Trninić, 1992), but the reviewer is right in that it does not affect the trend (i.e., regression slope against time). Also, it is routinely carried out in both hydrologic and climatologic research. The methods section has been amended accordingly.

6) The introduction should be improved. The introduction should clearly convey what has been found previously on annual streamflow trends in Europe? What is the gap in the current literature? How is this approached by this study? Please also check the logic of individual sentences and the subdivision of the introduction into paragraphs.

**REPLY:** We thank the reviewer for this suggestion also underlined in the short comment by Adriaan Teuling. In the revised version of the manuscript, the introduction includes a more complete summary of what has been found by recent studies on annual streamflow trends in Europe, what is missing in the current literature and in which way this study will fill the gap. The revised introduction also relies on a more logical paragraphing.

6) The explanation of streamflow trends by trends in air temperature and precipitation remains a bit vague and overlooks areas where it is probably not possible to explain streamflow trends with trends in air temperature or precipitation (such as positive streamflow trends in northern Spain). Some arguments need to be clarified e.g. it is not clear to me how groundwater or snowmelt effects would affect annual (and not only seasonal or monthly) streamflow.

**REPLY:** The discussion on groundwater and snowmelt roles has been improved, also specifying that it will rely on speculation and literature and not direct measure or testing of such variables. The cases in which the observed discrepancies between river discharge and weather series could be explained by based on logical and science-supported hypotheses using likely drivers, will be highlighted with their most relevant examples (eg Northern Spain).

**Detailed comments**

P1, L28-30: The logic of the sentence is not clear. There is no contrast between a lot of research and not finding uniform streamflow trends in Europe. When mentioning a lot of research that aimed at investigating streamflow trends in Europe, this should be backed up by some references and their main findings (e.g. Stahl et al., 2010, Stahl et al., 2012).

**REPLY:** The introduction, and in particular the review of past studies and their findings, has been deeply improved in the revised version of the manuscript. References has been added, including those suggested by the reviewer.

P2, L33-34: Did these studies also analyze changes in annual streamflow volume? What were the main findings? How did seasonal streamflow change?

**REPLY:** As for the previous comment, the review of past studies and their findings has been deeply improved in the revised version of the manuscript.

P2 L40-47: The section on potential drivers of the streamflow trends remains a bit vague. Are changes in river cross-sections or boat tourism relevant for annual streamflow volumes?

**REPLY:** Yes, if the shape of the river section is altered, or if flow itself is altered with recreational basin or locks for navigation. These sentences will be however moved to the Discussions to streamline the logical flow of the introduction. A missing refere to Vag et al. will be added in the Bibliography.

P4, L97: I would suggest to first clearly list the criteria for selecting catchments and then mention the final number of selected catchments at the end.

**REPLY:** The methods has bene amended accordingly – filtering criteria has been described in the methods, while the resulting number of catchments retained for analysis are reported in the Results.

P4, L101-102: You may use this in the introduction in order to emphasize your contribution in comparison to previous studies.

**REPLY:** Suggestion accepted, the sentence will be integrated in the introduction**

P4, L103-109: The description of the criteria for inclusion/exclusion from the database should be very clear. It is not very clear whether you aimed at including only near natural catchments. Did you check information from the data providers on human interventions such as changes in flow abstractions etc. (that would directly influence the trends)? Your database contains \_3900 series of 65-years data. It is a lot of work to visually scan daily data of all these series. Could you provide some detail on how this was achieved? Did you apply any automatic screening tests? How were inhomogeneities identified?

**REPLY:** Accepted - see reply to R2 comment 2 above.**

P5, L123ff: Why would it make any difference in terms of trend slope whether you calculate it on the original data or on the anomalies?

**REPLY:** Accepted - see reply to R2 comment 5 above.**

P5, L128: Delete "To homogenize the annual streamflow series", since dividing by catchment area canno

---

## Author Response (AR2)

**Authors' response**

**Editor**

**Editor**: Thank you for your detailed and complete reply to the reviews. Both Reviewers have now provided their feedback on your new version of the manuscript. Referee#1 recommends publish as is since all the issues have been satisfactorily addressed; however, Referee#2 identifies some issues that, despite being addressed in your reply, cannot be fully found in the revised text. Could you please check whether all the changes were implemented in the revised version? Please, provide in your reply specific points to identify the modifications.

**Reply**: We appreciate the efforts of the editor and the reviewers in revising the manuscript another time. We checked and further modified the manuscript following the last document of "authors reply". We apologize for any inconvenience.

**Editor**: I have checked some of the new texts, and changes, you provide in your reply and couldn't find them in the revised manuscript. The Referee #2 is just wondering whether all were finally implemented or whether some not definite version was finally upload. I kindly ask you to check this, and I suggest that you provide in the new reply each point in the text were the changes can now be found in the new document so that the Referee is not concerned about this.

**Reply**: We followed the suggestions of the Editor and prepared a new document of reply.

**Reviewer #1**

**Reviewer**: I appreciate the authors' effort in improving the manuscript addressing all the reviewers' comments.

**Reply**: We thank the Reviewer #1 for his/her support.

**Reviewer #2**

**Reviewer**: I stumbled across various discrepancies between the file with the responses to the referee's comments and the actual changes in the manuscript. It should be checked whether really the correct files have been uploaded. The authors have responded to the first two of my comments (concerning the lengths of the records analysed and the criteria for inclusion in the database). The authors now describe more clearly how they selected the streamflow series for their analysis. They now clearly state that 3485 stations were used in the analysis over the period 1950–2013. The new Fig. 1 is unfortunately hard to read, since each of the 3913 stations is represented by an individual line. This could be better represented by a figure that shows the number of gauges over time.

**Reply:** This representation of the data availability is commonly used in scientific literature as reported in Durocher et al. 2019 (Hydrological Processes. 2019; 33:1143‑1151). The figure allows to summarize the length of dataset in functions of the number of gauged station. In particular, it clearly displays the maximum period when the data are available. We tried to improve the quality of figure as far as possible.

[Figure]

*Figure 1. Data availability.*

The new figures 2 and 3 illustrate a data gap and a series with a step change but I think they are not necessary for the manuscript.

**Reply**: We accepted the suggestion of the Reviewer and removed both the figures.

In the responses to the referee's comments the authors agreed with many of the comments and suggested modifications. However, unfortunately, these modifications have apparently in the end not been implemented.

**Reply**: We apologize for the misunderstanding. We prepared a new document modifying the last response to reviewer comments and specified the changes in the manuscript. In particular, the comments of reviewers are reported in black, the first reply is in blue and the current replay (new reply) in green.

**Previous review: Reviewer #2**

The study analyses trends in annual streamflow over the period 1950-2015 in Europe. This is a relevant topic certainly within the scope of HESS. The study generally applies standard methods for trend analysis (Theil-Sen slope, Mann-Kendall test). The spatial patterns of the trends are compared to spatial patterns of air temperature and precipitation. The study extends previous work on observed streamflow trends in Europe by including a higher number of catchments, particularly in Portugal, Spain, France and Italy. This was possible through assembling the database of streamflow records from various sources. The results largely confirm

previous studies with dominant positive trends in northern Europe and dominant negative trends in the Mediterranean region.

**Reply:** We thank Rev#2 for her/his comments and suggestions, which allowed to improve the quality of the manuscript in this revised version.

**New Reply:** No comments to add.

**Main comments:**

**Reviewer**: Since the study states that records with missing data for more than two years were excluded from the database (L 107), I initially assumed that the calculated trends all relate to the period 1950-2015, which, looking at Fig. 2a, is apparently not the case.

**Reply:** Fig.2a was deleted because it created misunderstanding both in Rev#1 and #2. The original dataset included 3913 stations and after the checks on reliability, consistency and uniformity of series of data, 428 stations were discarded. The 63-year study period (from 1950 to 2013) has been chosen as the optimal threshold between maximizing series length and avoiding missing data, as shown in the following plot.

**New Reply:** We prefer to maintain Fig.1. Please, see the above comments.

**Reviewer**: This has of course a strong influence on the results and needs to be clarified. If the series length vary between catchments it will probably be more useful to analyze trends for different periods with nearly complete records, as the trends of course depend on the period analyzed (as discussed in the introduction).

**Reply:** We agree with the Rev#2 about the influence of length of series of data on trend identification. Dixon et al. 2006 coped with this problem by splitting the dataset in periods of different length, with a different number of stations for each period (see also Birsan et al. 2005). Nevertheless, one of the added values of our work was to consider a continuous dataset as large as possible over the entire study domain in order to evaluate spatial trends over European basins with a consistent sample size. It was the same approach proposed in the recent work by Durocher et al. (2019) where stations with missing data were discarded and a single time frame for all study domain was considered.

**New Reply:** We are firmly convinced that one of the added values of our work is to consider a continuous dataset as large as possible over the entire study domain in order to evaluate spatial trends over European basins with a consistent sample size.

We added a sentence at the end of Section 1.2 as follows:

"In the present study, we decided to maintain the integrity of the dataset focusing on the same time frame for all the study domain without splitting it in periods of different lengths. This procedure was already proposed in the study of Durocher et al. (2019) where preferred to discard all those time-series with missing data over a threshold rather than considered different time windows." (lines 190-195)

**Reviewer**: The criteria for inclusion/exclusion from the database should be described very clearly. It is not so clear whether the study aimed at only including near natural catchments. How were gaps smaller than 2

years treated? The steps that were undertaken to exclude inhomogeneous series, or series strongly affected by human interventions need to be mentioned clearly. For example, did the authors try to get information from the data providers on human interventions such as changes in flow abstractions etc. It should be described clearly how the database was 'consolidated and validated'. Did you apply any automatic screening tests to systematically check the series for possible inhomogeneities?

**Reply:** We thank the reviewer for raising this issue. In the revised manuscript, we have added details about the pre-processing activity done to select the discharge time series used for the analysis. In particular, to ensure quality of discharge observations, the following steps were followed: 1) check on data availability; 2) check for outliers (i.e. five st.dev. higher or lower than the means; 3) check on the presence of in-homogeneities through automatic screening tests. In order to filter out catchments affected by human disturbance, each discharge time series was accurately scrutinized through visual hydrograph inspection to identify disturbed hydrographs due to e.g. the presence of dams/reservoirs. Discharge time series characterized by disturbed hydrographs were discarded from the analysis. It should be noted that most of the basins considered in the analysis are taken from the EWA database, i.e. a discharge data collection of near-natural streamflow records from small catchments (Stahl et al, 2010). Moreover, the Global Reservoir and Dam (GRanD, https://sedac.ciesin.columbia.edu/data/set/grand-v1-dams-rev01) has been used to identify if (how many) dams/reservoirs are actually present in the selected basins. At the end of this analysis we expect that no substantial differences will be found between the basins retained for the analysis and the basins for which a certain degree of disturbance can be tolerated. Only stations with low human impact (no presence of dams/reservoir in the analysis period or no appreciable dam impact in the hydrograph); with less than 20% of missing data, showing no inhomogeneities in the time series were retained in the compiled dataset. Gaps smaller than two years were retained as missing data; during trend calculations, missing data were discarded on a case-by-case basis.

**New Reply:** We added two bullets in the methodology regarding this time-series check:

"The quality control was conducted in succession on daily and aggregated time-series following the steps reported in Gudmundsson and Seneviratne (2016):

(i)     a visual hydrograph inspection to identify evident malfunction, consistent gaps (Fig. 2) and hydrograph disturbs such as presence of dams or reservoirs;

(ii)    excluding catchments with a drainage area larger than 100,000 km$^2$ to minimize the possibility that the human actives can significantly cause disturbances on the streamflow time-series (Piniewski et al. 2018);

(iii)   remove values with negative daily streamflow values;

(iv)    remove time-series with more than 2 years of missing data."

See lines 120-135

**Reviewer**: Some results are not very clear. The results section reports significant trends in 95% of the stations, which disagrees with results reported in Table 1. In the results section, it is not always clear whether results on trends also include non-significant trends.

**Reply:** The number of basins reported in tab 1 (tab. 2 in the revised version of the manuscript) were incorrectly transcribed by the authors. They referred to the total number of stations in each macro-region (i.e. 3,485). In the table, only significant positive or negative trends are shown. These were 95% of total gauged stations (i.e. 3,310 stations). In the revised manuscript, the number of stations in each macro-region has been corrected. The manuscript will also clearly state whether any summary result includes non-significant trends.

**New Reply**: We added into the manuscript:

We clarified the results modifying Tab. 2.

"Results found that in 95% of the European gauged stations (i.e. 3,310 stations) the MK test confirmed the presence of a trend in annual streamflow volumes."

In addition, we modified Table 2:

Tab. 2. Number of significant (i.e. 3,310 stations) positive and negative trends in annual streamflow volumes in the European macro-regions.

| Region | Number of stations | Positive trend | Negative trend |
|---|---|---|---|
| Boreal | 323 | 307 | 16 |
| Continental | 694 | 472 | 222 |
| Atlantic | 1191 | 846 | 345 |
| Mediterranean | 1102 | 88 | 1014 |
| Total | 3310 | 1713 | 1597 |

**Reviewer**: I disagree with the finding of an inversion point in 1985 for the average series in the Mediterranean region. I do not see a change in the trend direction or trend slope in 1985. The fact that streamflow is above average before and below average after 1985 is a rather arbitrary result that depends on the selected study period. Streamflow has been decreasing since about 1965, and if anything, the rate of decrease has rather slowed down since the late 1980s.

**Reply:** The reviewer is right. Figure 7 in the manuscript highlights that streamflow has been decreasing since about 1965 and the rate of decrease has rather slowed down since the late 1980s. In the revised manuscript the sentences related to Figure 7 has been modified accordingly and supported by new statistical trend analyses on the entire time period.

*"Fig. 7 shows a change in the annual streamflow volume pattern between 1980 and 1985 moving from positive to negative availabilities with respect to the mean of annual streamflow volume observations. This finding is consistent with the results found by Hannaford et al. (2013) on the marked decreasing of low flow regimes in southern Europe in the last thirty years as well as with the conclusions of the International Panel*

*of Climate Change (IPCC) work on climate change prospective (IPPC 2007) which highlighted how in the Northern Hemisphere climate change effects in reducing water resource availability have increased notably from the post- 1980 period."*

**New Reply:** This part has been already added in the manuscript.

**Reviewer**: The calculation of the Sen's slope from annual streamflow anomalies is described as innovative, but if I do not overlook something this should not affect trends (and has probably been done in many studies).

**Reply:** By using anomalies to detect trends, the absolute random error is minimized (Pandžić and Trninić, 1992), but the reviewer is right in that it does not affect the trend (i.e., regression slope against time). Also, it is routinely carried out in both hydrologic and climatologic research. The methods section has been amended accordingly.

**New Reply:** The term "innovative" was removed. We modified as follows:

"Theil-Sen's line, known as Theil-Sen's slope or Sen's slope, was calculated on the annual anomalies in streamflow volumes, an alternative modality with respect to the common application on direct streamflow data (Birsan et al. 2005). "

**Reviewer**: The introduction should be improved. The introduction should clearly convey what has been found previously on annual streamflow trends in Europe? What is the gap in the current literature? How is this approached by this study? Please also check the logic of individual sentences and the subdivision of the introduction into paragraphs.

**Reply:** We thank the reviewer for this suggestion also underlined in the short comment by Adriaan Teuling. In the revised version of the manuscript, the introduction includes a more complete summary of what has been found by recent studies on annual streamflow trends in Europe, what is missing in the current literature and in which way this study will fill the gap. The revised introduction also relies on a more logical paragraphing.

**New Reply:** We largely modified the introduction summarizing the shared results on the trend detection as follows:

[revised manuscript text omitted]

**Reviewer**: The explanation of streamflow trends by trends in air temperature and precipitation remains a bit vague and overlooks areas where it is probably not possible to explain streamflow trends with trends in air

temperature or precipitation (such as positive streamflow trends in northern Spain). Some arguments need to be clarified, e.g., it is not clear to me how groundwater or snowmelt effects would affect annual (and not only seasonal or monthly) streamflow.

**Reply:** The discussion on groundwater and snowmelt roles has been improved, also specifying that it will rely on speculation and literature and not direct measure or testing of such variables. The cases, in which the observed discrepancies between river discharge and weather series, could be explained by based on logical and science-supported hypotheses using likely drivers, will be highlighted with their most relevant examples (eg Northern Spain).

**New Reply:** We improve the discussion as follow:

"Concerning air temperature changes, the works of Staggle et al. (2017), Vicente-Serrano et al. (2014), Spinoni et al. (2015), Zeng et al. (2012), Willems (2013) and Madsen et al. (2014) confirm a global increase of mean temperatures with a marked trend in Mediterranean areas, where air temperature is expected to increase up to 0.3 °C/decade (as found in this study). The increase of air temperature directly impact glacierized and snow dominated basins where it can be responsible of the increase of runoff volume during the last sixty years due to the loss of ice masses (Sommer et al. 2020), however, depending on the basin elevation and trend in precipitation, some glaciers might have lost some sensitivity to an increased runoff production as a consequence of higher temperatures since there has not been more ice to melt and because, at high elevations, temperature might be not warm enough to counter balance the precipitation trend. In summary, for glacierized basins (or that use to be) there might be a causal effect of temperature on increased runoff volume (although this effect might have lost in time for some of them as explained below) while, for the others, precipitation seems again the main driver of runoff trend as it can be seen over the Alps by the contrasting trend found between the Italian side (negative) and continental side (positive) which reflects the trend in precipitation. On the other hand, temperature increase can impact negatively runoff over energy-limited environments by increasing evapotranspiration (Teuling et al. 2013, Avanzi et al. 2020) so some catchments might have experienced reduced runoff trend as a consequence of warming. This might explain the negative runoff trend found for some basins at high latitudes." (lines 305-340)

**Detailed comments**

**Reviewer**: P1, L28-30: The logic of the sentence is not clear. There is no contrast between a lot of research and not finding uniform streamflow trends in Europe. When mentioning a lot of research that aimed at investigating streamflow trends in Europe, this should be backed up by some references and their main findings (e.g. Stahl et al., 2010, Stahl et al., 2012).

**Reply:** The introduction, and in particular the review of past studies and their findings, has been deeply improved in the revised version of the manuscript. References has been added, including those suggested by the reviewer.

**New Reply**: We included additional references clarifying their outcome and findings as follows:

"Although the hydrological scientific community undertook a great effort, few research robustly demonstrates an ubiquitous and uniform trend in European annual streamflow volumes (e.g., Mediero et al., 2015; Alfieri et al., 2015; Hodgkins et al. 2017; Blöschl et al. 2019). ..."

Then, we largely modified the introduction as shown in one of the previous point.

**Reviewer**: P2, L33-34: Did these studies also analyze changes in annual streamflow volume? What were the main findings? How did seasonal streamflow change?

**Reply:** As for the previous comment, the review of past studies and their findings has been deeply improved in the revised version of the manuscript.

**New Reply:** We largely modified the introduction as shown in one of the previous point.

**Reviewer**: P2 L40-47: The section on potential drivers of the streamflow trends remains a bit vague. Are changes in river cross-sections or boat tourism relevant for annual streamflow volumes?

**Reply:** Yes, if the shape of the river section is altered, or if flow itself is altered with recreational basin or locks for navigation. These sentences will be however moved to the Discussions to streamline the logical flow of the introduction. A missing reference to Vag et al. will be added in the References.

**New Reply:** We delete this part in the new version of the manuscript

**Reviewer**: P4, L97: I would suggest to first clearly list the criteria for selecting catchments and then mention the final number of selected catchments at the end.

**Reply:** The methods has bene amended accordingly – filtering criteria has been described in the methods, while the resulting number of catchments retained for analysis are reported in the Results.

**New Reply:** We largely modified the materials and methods, as follows:

"For assessing the reliability of streamflow daily values of each gauged station of the original dataset, a quality control and a homogeneity assessment were performed according the methodologies described in Buishand (1984), Chu et al. (2014), Ghiggi et al. (2019) and Kundzewicz (2015).

The quality control was conducted in succession on daily and aggregated time-series following the steps reported in Gudmundsson and Seneviratne (2016):

(v)     a visual hydrograph inspection to identify evident malfunction, consistent gaps (Fig. 2) and hydrograph disturbs such as presence of dams or reservoirs;

(vi)    excluding catchments with a drainage area larger than 100,000 $km^2$ to minimize the possibility that the human actives can significantly cause disturbances on the streamflow time-series (Piniewski et al. 2018);

(vii)   remove values with negative daily streamflow values;

(viii)     remove time-series with more than 2 years of missing data.

The homogeneity detection of data series (Fig. 3) was performed combining four different tests (Gudmundsson et al. 2018): (i) the standard normal homogeneity test of Alexandersson (1986); (ii) the Buishand range test (Buishand, 1982); (iii) the Pettitt test (Pettitt, 1979) and (iv) the Von Neumann ratio test (von Neumann, 1941). Homogeneity tests were carried out using the "iki.dataclim" statistical package for R (Orlowsky, 2014). The streamflow time series were considered as consistent when the null hypothesis at the 1% level was accepted at least in 3 of 4 tests (ECA&D) (Gudmundsson and Seneviratne, 2016; Merino et al., 2016). Despite potential levels of human-induced alterations of river flow regime could be still present in time-series data after the application of the aforementioned controls, a certain degree of disturbance can be tolerated (Murphy et al. 2013). In order to further reduce the disturbance, high flow conditions were not investigated and we focused the analysis on annual streamflow volumes."

**Reviewer**: P4, L101-102: You may use this in the introduction in order to emphasize your contribution in comparison to previous studies.

**Reply**: Suggestion accepted, the sentence will be integrated in the introduction.

**New Reply**: We added a key point into the objective as suggested:

"and (iv) to discuss the outcomes of the present study with previous investigations."

**Reviewer**: P4, L103-109: The description of the criteria for inclusion/exclusion from the database should be very clear. It is not very clear whether you aimed at including only near natural catchments. Did you check information from the data providers on human interventions such as changes in flow abstractions etc. (that would directly influence the trends)? Your database contains 3900 series of 65-years data. It is a lot of work to visually scan daily data of all these series. Could you provide some detail on how this was achieved? Did you apply any automatic screening tests? How were inhomogeneities identified?

**Reply**: Accepted - see reply to R2 comment 2 above.

**New Reply**: We largely modified the materials and methods (please see par 2.1 in the current version of the manuscript)

**Reviewer**: P5, L123ff: Why would it make any difference in terms of trend slope whether you calculate it on the original data or on the anomalies?

**Reply**: Accepted - see reply to R2 comment 5 above.

**New Reply**: We modified in the current version of the manuscript (see par. 2.1).

**Reviewer**: P5, L128: Delete "To homogenize the annual streamflow series", since dividing by catchment area cannot homogenize a time series.

**Reply**: Deleted.

**New Reply:** Done.

**Reviewer**: P5, L132ff: Have you checked the streamflow series for autocorrelation? How did you deal with series that contain significant autocorrelation?

**Reply:** The streamflow series of data were checked with lag-1 autocorrelation coefficient as proposed by Khaliq et al. (2009). The autocorrelation levels are reported in the picture in response to comment 4 of Rev#1. No series was significantly autocorrelated.

**New Reply:** We inserted in the manuscript the following section:

"About 90% of stations belongs to catchments with size less than 1,000 km2 of which more than 50% ranging from 1 to 200 km$^2$. Temporal autocorrelation level of the selected near-natural daily streamflow series was verified calculating lag-1 serial autocorrelation coefficient with a 95% of confidence bounds as suggested by Khaliq et al. (2009), Kulkarni and von Storch (1995) and von Storch (1995). All autocorrelation coefficients were found included in the confidence bounds, as shown in Fig. 5, and, therefore, they can be considered ready for the trend identification."

[Figure]

**Fig.5. Samples autocorrelations. Red points are the value of lag-1 autocorrelation coefficient, whereas black dotted lines represent the 95% confidence bounds.**

**Reviewer**: P6, L138: Since the streamflow volumes were divided by area, runoff depths would be more appropriate (instead of streamflow volume), no? (adjust throughout the paper)

**Reply:** Suggestion rejected – the reviewer is right, but streamflow volume is a widespread measure, which is readily understandable by managers and citizens. We decided to keep it that way.

**New Reply:** We decided to keep it that way.

**Reviewer**: P6, L145 and 146: This seems not correct, Table 1 shows positive trends in 7% and negative trends in 5% of the catchments?

**Reply:** The overall figures were corrected – 52% of positive trends and 48%, consistent with Table 2.

**New Reply**: We modified Tab. 2.

**Reviewer**: P6 Fig. 3: These figures are not necessary in my opinion.

**Reply:** The figure has been left in the revised manuscript just as an example of trend calculation.

**New Reply:** We decided to keep it that way.

1.  **Reviewer**: P6, L151: The unit of annual streamflow per area is length/time (e.g. $m^3/(km^2\ year)$, or $mm\ y^{-1}$). Therefore the change in runoff over a certain period is $length/time^2$ (e.g. $m^3/(km^2\ year^2)$).

**Reply**: Accepted – the values and units will be updated to reflect yearly change expressed in m3/(km2 year2).

**New Reply:** Done.

**Reviewer**: P7, L170; legend and caption of Fig. 5: replace rainfall by precipitation (assuming that snow is included).

**Reply**: Snow is included. Suggestion accepted.

**New Reply:** Done. We modified the caption as follows:

"Fig. 8. Comparison between annual streamflow volume trends and daily mean temperature (a) and rainfall (included snow-to-liquid equivalent) (b) trends over the European continent. Only significant trend are shown."

---

## Author Response (AR3)

**Authors' response**

**Editor**

**Editor**: Thank you for your reply and revised version following the review by Referee #2. However, following the new reply by Referee#2 and my own reading, in many cases the modifications you indicate in your reply are not implemented in the revised text (nor in the tracked-changes document). Could you please check again carefully whether all the changes were implemented in the revised version, or whether a previous version of the manuscript was wrongly uploaded? Referee#2 provides some examples of these mismatches, but more cases can be identified along the text. For further clarification, the current versions you uploaded are named as follows:

hess-2020-21-ATC1.pdf
hess-2020-21-manuscript-version4.pdf
hess-2020-21-author_response-version2.pdf

Looking forward to receiving your news, I apologize for my delay, due to health issues in my family that have expanded longer than expected. I am sorry for the inconveniences.

**Reply**: We are sorry for the misunderstandings. We made some efforts to reply each comment of the Reviewer #2 clarifying the aspects.

**Reviewer #2**

**Reviewer**: 1) The authors analysed trends over one common time period, 1950-2013, and allowed a maximum of two years missing data. 3485 stations of their original 3913 stations fulfilled this criterion (and further quality criteria). Unfortunately, they decided to keep the new Fig. 1 instead of a figure that shows the number of available stations over time (similar to the previous Fig. 2a). In the previous Fig. 2a the number of stations during the 1950s is around 1000-1500 and always smaller than 2500 during the studied period. It is unclear how 3485 stations could be found that have less than two years missing between 1950-2013. Could the authors please clarify? If the series have different length, this needs to be described very clearly in the methods and be taken into account for the interpretation of the results.

**Reply**: As we already answered in the second round of revision, Fig.2a was deleted because it created misunderstanding both in Rev#1 and #2. The original dataset included 3913 stations and after the checks on reliability, consistency and uniformity of series of data, 428 stations were discarded. The 63-year study period (from 1950 to 2013) has been chosen as the optimal threshold between maximizing series length and avoiding missing data. We agree with the Reviver about the influence of length of series of data on trend identification. As already we answered, Dixon et al. 2006 coped with this problem by splitting the dataset in time frames of different length, with a different number of stations for each period (see also Birsan et al. 2005). Nevertheless, one of the added values of our work was to consider a continuous dataset as large as possible over the entire study domain in order to evaluate spatial trends over European basins with a consistent sample size. It was the same approach proposed in the recent work by Durocher et al. (2019) where stations with long series of missing data were discarded and a single time frame for all study domain was considered.

**Reviewer**: 2) The authors have amended their discussion on the causes of the streamflow trends. In their discussion on precipitation changes as a driver they state "Concerning rainfall changes, the southern regions are affected by a marked negative trend (even below -3 mm/decade), while the northern regions are characterized by a positive trend which can overcome 10 mm/decade. The spatial distribution over the 230 continent of both patterns appears perfectly congruent with the findings in annual streamflow volumes, as shown in Fig.6." I still strongly disagree with "perfectly congruent" changes in spatial patterns of streamflow

and precipitation changes. For example over Germany, precipitation has largely slightly decreased while discharge has increased. What could be the possible reasons for this pattern? This needs to be discussed in the manuscript.

**Reply**: Here, we referred to "Mediterranean" and "Atlantic-Boreal" areas for "Southern" and" Northern" regions, respectively. It seems clear that there are transition or intermediate areas in which these trends are not marked as the Central part of Germany. Thus, we clarified modifying the sentence as follows:

"*Concerning precipitation changes, the Mediterranean regions are affected by a marked negative trend (even below -3 mm/decade), while Boreal and Atlantic regions are characterized by a positive trend which can overcome 10 mm/decade.*"

**Reviewer**: 3) While the authors already agreed with many of the referee comments in their previous version of their responses to the referee's comments, these changes could not be found in their revision 1 text. In the second revision, more of the changes that are described in the author responses are actually implemented in the text. However, this does not apply to all changes and it is annoying to see that the authors state in two rounds of revisions in their replies that changes were undertaken that in the end cannot be found in the text. Is it the reviewer's responsibility to check one by one that authors are not only pretending to undertake changes? I ask the authors to go very carefully through their replies and check whether really all of them are also in the text. Here are examples that I could not find in the text:

**Reply**: We are sorry for the misunderstandings. We carefully check the last version.

Reviewer: The calculation of the Sen's slope from annual streamflow anomalies is described as innovative, but if I do not overlook something this should not affect trends (and has probably been done in many studies).

Reply: By using anomalies to detect trends, the absolute random error is minimized (Pandžić and Trninić, 1992), but the reviewer is right in that it does not affect the trend (i.e., regression slope against time). Also, it is routinely carried out in both hydrologic and climatologic research. The methods section has been amended accordingly.

New Reply: The term "innovative" was removed. We modified as follows:

"Theil-Sen's line, known as Theil-Sen's slope or Sen's slope, was calculated on the annual anomalies in streamflow volumes, an alternative modality with respect to the common application on direct streamflow data (Birsan et al. 2005). "

=> Not implemented in the text.

**Reply**: We inserted the sentence in red inside the manuscript with corrections.

Reviewer: P6, L151: The unit of annual streamflow per area is length/time (e.g. $m3/(km2\ year)$, or $mm\ y-1$). Therefore the change in runoff over a certain period is length/time2 (e.g. $m3/(km2\ year2)$).

Reply: Accepted – the values and units will be updated to reflect yearly change expressed in $m3/(km2\ year2)$.

New Reply: Done.

=> Not implemented in the text.

**Reply**: Probably, there was a misunderstood. In our work we refer to 'annual streamflow volume' as declared immediately in the abstract. The trend of this variable over the years is expressed in volume/time. Therefore, the annual streamflow volume per area is expressed in volume/(time x surface).

Reviewer: P7, L170; legend and caption of Fig. 5: replace rainfall by precipitation (assuming that snow is included).

Reply: Snow is included. Suggestion accepted.

New Reply: Done. We modified the caption as follows:

"Fig. 8. Comparison between annual streamflow volume trends and daily mean temperature (a) and rainfall (included snow-to-liquid equivalent) (b) trends over the European continent. Only significant trend are shown."

=>Not implemented in the text.

**Reply**: We inserted the sentence in red inside the manuscript with corrections.

Please also replace rainfall with precipitation in the main text.

**Reply**: We replaced "rainfall" with "precipitation".

---

## Author Response (AR4)

**Reply document**

Dear prof. María J. Polo,

We want to thank for the efforts, time and patience in following the revision of this manuscript.

We followed your two critical points to complete definitively the work.

*1. From the reading of the manuscript, I can conclude that a common time window for the analysis was chosen, that is the period from 1950 to 1963, and that all stations with more than 2 years of gaps in the time series for such period were removed (i.e., the final 3485 stations provide at least 61 years of flow records). Could you confirm this is correct? If this was not the case, please, could you state clearly in the text (section 2.1) that, despite a common time window in your analysis for the flow time series, not all the stations provide full-length series during such period, and add a short paragraph in the discussion commenting on the potential consequences of this on the interpretation of results?*

We confirm the interpretation. Then, we inserted the statement into section 2.1 as suggested.

*"The application of quality control and homogeneity tests led to discard 428 series of data. Thus, 3,485 stations providing at least 61 years of flow records, were selected and assembled into a dataset that guarantees the best balance between the necessities to investigate a dataset as large as possible (which covers a large part of the continent and a nearly complete period of analysis), and to detect a historical variability."*

Moreover, we inserted a comment for discussing the consequences of this choice.

*"Results found that in 95% of the European gauged stations (i.e., 3,310 stations) the MK test confirmed the presence of a trend in annual streamflow volumes. In general, 70% of positive and 30% of negative trends in annual streamflow volume anomalies is recognized, with clear positive trend in northern regions and negative trend in southern ones, as shown in Fig. 5. These results are certainty representative of the selected time window, despite not all gauged stations provide full-length series during the same period, because the data selection leads to maintain a trade-off between record length and spatial coverage over the continent, meantime to remove those limited by wide time-series gap or evident uncertainties/inconsistencies."*

*2. Despite it can be assumed from the text that you are analyzing annual streamflow per area, it is more accurate to use volume/area/(time.time) as the units of the change of this variable over time. Please, I strongly recommend to use these units instead of just volume/area/time.*

We accepted the suggestions and modified the unit of measure in the entire manuscript and in all figures.